# Autophagy regulates lipid metabolism through selective turnover of NCoR1

Tetsuya Saito[1], Akiko Kuma[2,3,4], Yuki Sugiura[4,5], Yoshinobu Ichimura[1], Miki Obata[1], Hiroshi Kitamura[6], Shujiro Okuda [7], Hyeon-Cheol Lee[8], Kazutaka Ikeda[9], Yumi Kanegae[10], Izumu Saito[11,16], Johan Auwerx[12], Hozumi Motohashi [6], Makoto Suematsu [5], Tomoyoshi Soga[13], Takehiko Yokomizo[8], Satoshi Waguri[14], Noboru Mizushima [2] & Masaaki Komatsu[1,15]

Selective autophagy ensures the removal of specific soluble proteins, protein aggregates, damaged mitochondria, and invasive bacteria from cells. Defective autophagy has been directly linked to metabolic disorders. However how selective autophagy regulates metabolism remains largely uncharacterized. Here we show that a deficiency in selective autophagy is associated with suppression of lipid oxidation. Hepatic loss of *Atg7* or *Atg5* significantly impairs the production of ketone bodies upon fasting, due to decreased expression of enzymes involved in β-oxidation following suppression of transactivation by PPARα. Mechanistically, nuclear receptor co-repressor 1 (NCoR1), which interacts with PPARα to suppress its transactivation, binds to the autophagosomal GABARAP family proteins and is degraded by autophagy. Consequently, loss of autophagy causes accumulation of NCoR1, suppressing PPARα activity and resulting in impaired lipid oxidation. These results suggest that autophagy contributes to PPARα activation upon fasting by promoting degradation of NCoR1 and thus regulates β-oxidation and ketone bodies production.

[1] Department of Biochemistry, Niigata University Graduate School of Medical and Dental Sciences, Chuo-ku, Niigata 951-8510, Japan. [2] Department of Biochemistry and Molecular Biology, Graduate School and Faculty of Medicine, The University of Tokyo, Bunkyo-ku, Tokyo 113-0033, Japan. [3] Department of Genetics, Graduate School of Medicine, Osaka University, Suita, Osaka 565-0871, Japan. [4] Japan Science and Technology Agency, PRESTO, Saitama 332-0012, Japan. [5] Department of Biochemistry, Keio University School of Medicine, Tokyo 160-8582, Japan. [6] Department of Gene Expression Regulation, Institute of Development, Aging and Cancer, Tohoku University, Sendai 980-8575, Japan. [7] Bioinformatics Laboratory, Niigata University Graduate School of Medical and Dental Sciences, Chuo-ku, Niigata 951-8510, Japan. [8] Department of Biochemistry, Juntendo University Graduate School of Medicine, Bunkyo-ku, Tokyo 113-8421, Japan. [9] Laboratory for Metabolomics, RIKEN Center for Integrative Medical Sciences (IMS), Yokohama, Kanagawa 230-0045, Japan. [10] Core Research Facilities of Basic Science (Molecular Genetics), Research Center for Medical Science, Jikei University School of Medicine, Tokyo 105-8461, Japan. [11] Laboratory of Molecular Genetics, Institute of Medical Science, The University of Tokyo, Tokyo 108-8639, Japan. [12] Laboratory of Integrative and Systems Physiology, École Polytechnique Fédérale de Lausanne (EPFL), 1015 Lausanne, Switzerland. [13] Institute for Advanced Biosciences, Keio University, Tsuruoka, Yamagata 997-0052, Japan. [14] Department of Anatomy and Histology, Fukushima Medical University School of Medicine, Hikarigaoka, Fukushima 960-1295, Japan. [15] Department of Physiology, Juntendo University Graduate School of Medicine, Bunkyo-ku, Tokyo 113-8421, Japan. [16]Present address: Laboratory of Virology, Institute of Microbial Chemistry, Shinagawa-ku, Tokyo 141-0021, Japan. Correspondence and requests for materials should be addressed to M.K. (email: mkomatsu@juntendo.ac.jp)

Autophagy is an intracellular protein degradation pathway mediated by lysosomes. Among several known autophagic pathways, the best understood is macroautophagy (hereafter referred to as autophagy), which is characterized by formation of a double-membrane structure called the autophagosome that fuses with the lysosome. The lysosome contains various hydrolases, including proteases, lipases, glycosidases, and nucleases; therefore, autophagy can degrade multiple types of cytoplasmic components simultaneously and provide the resultant molecular building blocks, such as amino acids, glucose, nucleotides, and fatty acids, for use by starving cells[1]. Experiments in mouse models have shown that autophagy is required to maintain the levels of amino acids and glucose in blood and tissues of neonatal and adult mice during fasting[2–6]. Therefore, in order to adapt to fasting, mice require the supply of nutrient molecules provided by autophagic degradation.

Upon fasting, cells shift their metabolism from glucose metabolism to fatty-acid oxidation to produce energy[7,8]. For fatty-acid oxidation to occur, triglycerides stored in the cell as lipid droplets must be degraded to fatty acids in the cytosol and/or lysosomes. Hydrolysis of triglycerides by cytoplasmic neutral lipases, such as adipose triglyceride lipase, hormone sensitive lipase, and carboxylesterases, are the major mechanisms for mobilizing triglycerides into fatty acids[8,9]. Lysosomes are also involved in hydrolysis of triglycerides. In fact, ablation of lysosomal acid lipase in mice results in massive triglyceride and cholesterol storage in the liver[10]. Selective autophagic degradation of lipid droplets in lysosomes (termed lipophagy) is thought to contribute to mobilization of triglycerides during starvation[11]. Supply of lipids through canonical autophagy, but not lipophagy, is also required to replenish triglycerides in lipid droplets, which are the source of molecules for fatty-acid oxidation[12].

Since the discovery of lipophagy in mammals, many studies have shown that autophagy is involved in lipid metabolism, including lipogenesis, lipolysis, fatty-acid oxidation, ketogenesis, and cholesterol efflux[13], and that loss of autophagy in the liver obstructs the mobilization of triglycerides into fatty acids, resulting in steatosis and an insulin-resistant state[11,14]. The physiological importance of lipophagy has been described in neurons, macrophages, cancer cells, and enterocytes[15–18]. Recently, however, conflicting results regarding lipophagy have been reported. First, although the accumulation of lipid droplets in autophagy-deficient mouse livers is believed to constitute conclusive evidence of lipophagy, several independent analyses of knockout mice lacking autophagic components specifically in hepatocytes have observed a reduction in the number of lipid droplets[19–23]. Second, the autophagy machinery participates in lipid droplet formation in hepatocytes and cardiomyocytes[22,24], and deletion of autophagy-related genes in mouse liver decreases the level of triglycerides[22] and impairs ketogenesis[23]. Third, suppression of autophagy represses the activity of the nuclear receptor, liver X receptor α (LXRα), which has a vital role in fatty-acid synthesis, and prevents liver steatosis under physiological fasting and high-fat diet conditions[21]. Therefore, it is plausible that the role of autophagy in lipid metabolism is beyond the scope of simple degradation of lipid droplets and membranes, implying that lipid metabolism is regulated by autophagy in an as-yet-uncharacterized way.

Autophagy can regulate transactivation of a transcription factor through selective turnover of its corresponding regulator. Nuclear factor erythroid 2-related factor 2 (Nrf2) induces a battery of genes encoding anti-oxidant proteins and proteins related to proteostasis, such as proteasome subunits and autophagy-related proteins. p62/SQSTM1 (henceforth p62) interacts with the autophagosome-localizing protein LC3 through its LC3-interacting region (LIR) and is degraded by autophagy[25–27].

p62 also binds to Kelch-like ECH-associated protein 1 (Keap1), an adaptor protein of the Cullin-3-based ubiquitin ligase for Nrf2, and inhibits its E3 activity[28,29]. Thus, quantitative regulation of p62 via autophagy determines Nrf2 activity. In fact, loss of autophagy in mouse liver is accompanied by persistent activation of Nrf2 due to prominent accumulation of p62, leading to liver enlargement, severe liver injury, and benign liver adenoma, all of which are suppressed by concomitant loss of p62 or Nrf2[30–32].

In this study, we identify a form of regulation of lipid metabolism through selective autophagy of NCoR1, a negative regulator of the nuclear receptor PPARα. Inactivation of autophagy in mouse livers is accompanied by accumulation of NCoR1 and subsequent inactivation of PPARα, resulting in impaired lipid oxidation and reduced production of ketone bodies under fasting conditions. In mechanistic terms, NCoR1 interacts with gamma-aminobutyric acid receptor-associated protein (GABARAP), an autophagosome-localizing protein, in a GABARAP-interacting motif (GIM)-dependent manner, and is degraded by autophagy. Our data provide insights into the physiological role of selective autophagy in metabolic regulation.

## Results

**Impaired lipid oxidation in autophagy-deficient livers.** To investigate the changes in lipid metabolism resulting from the loss of autophagy, we conducted comprehensive lipidome analyses in livers of control ($Atg7^{f/f}$) and mutant mice carrying a hepatocyte-specific knockout of $Atg7$, a gene essential for autophagy ($Atg7^{f/f}$;Alb-*Cre* mice). In the livers of 5-week-old $Atg7^{f/f}$ and $Atg7^{f/f}$;Alb-*Cre* mice, we identified a total of 524 individual lipid species by liquid chromatography–tandem mass spectrometry (LC–MS/MS) (Supplementary Figure 1). Among them, acylcarnitine molecular species differed markedly between the two groups (Supplementary Figure 2). Carnitine is a carrier for long-fatty acids, and facilitates their transport into mitochondria for lipid oxidation, which is markedly induced in parallel with autophagy (Fig. 1a). Although the levels of most acylcarnitine species in liver were significantly higher in $Atg7^{f/f}$;Alb-*Cre* mice than in $Atg7^{f/f}$ mice at 5 weeks of age, under both fed and fasting conditions (Fig. 1b), the amount of carnitine was lower under fasting conditions (Fig. 1c). Similar alterations were also observed in livers of $Atg5^{f/f}$; Mx1-*Cre* mice 1 to 2 weeks after intraperitoneal injection of polyinosinic-polycytidylic acid (pIpC), which induced liver-specific deletion of $Atg5$, another gene essential for autophagy (Fig. 1b and c). We did not observe any morphological abnormal mitochondria in hepatocytes of both 5-week-old $Atg7^{f/f}$;Alb-*Cre* mice and of $Atg5^{f/f}$;Mx1-*Cre* mice at 1 to 2 weeks after injection of pIpC (Supplementary Figure 3). Considering that the failure of mitochondrial β-oxidation increases the ratio of acylcarnitine/carnitine[33], these results suggest that lipid metabolism is impaired by loss of autophagy at the step of β-oxidation, rather than at the step of fatty-acid supply. Indeed, while blood glucose and free fatty-acid levels in $Atg7^{f/f}$;Alb-*Cre* and pIpC-injected $Atg5^{f/f}$;Mx1-*Cre* mice were comparable with those in control mice (Fig. 1d, e), the blood level of β-hydroxybutyrate (β-OHB), one of the ketone bodies, was significantly lower in both mutant mice than in control mice under fasting conditions (Fig. 1f), suggesting that lipid oxidation was suppressed in the livers of both types of mutant mice. In fact, we observed lower levels of β-OHB in the liver in pIpC-injected $Atg5^{f/f}$;Mx1-*Cre* mice than in control mice, regardless of nutrient conditions (Fig. 1g, left graph). An in vivo tracer study using [$^{13}$C]-palmitate revealed that fasting produces $^{13}$C-β-OHB, but the amount in mutant livers was much lower (Fig. 1g, right graph). These results suggest that β-oxidation is impaired in autophagy-deficient livers, resulting in lower production of ketone bodies under fasting conditions.

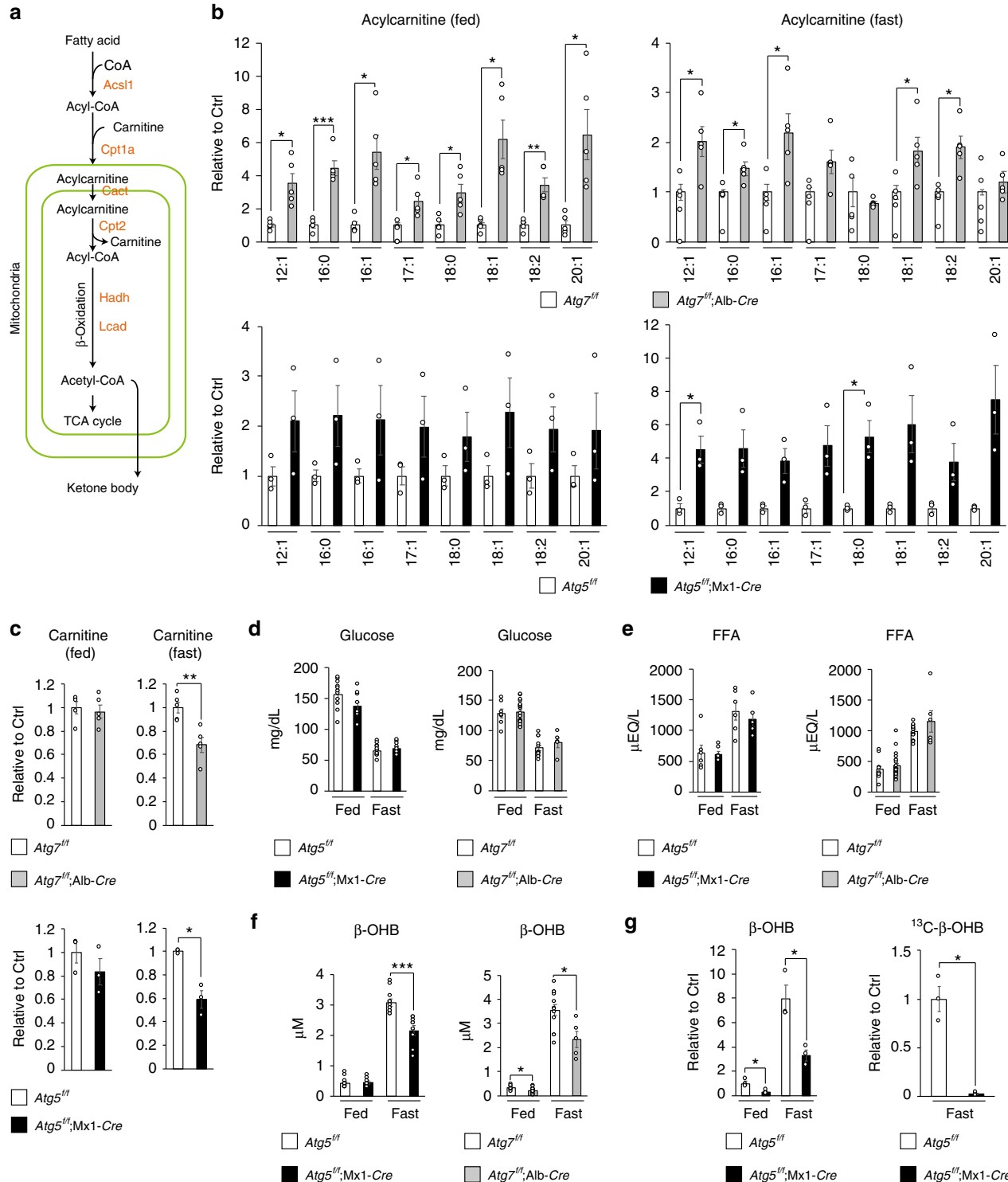

**Fig. 1** Decreased lipid-oxidation in autophagy-deficient livers. **a** Enzymes involved in lipid catabolism, and representative metabolites produced by the enzymes. **b** Levels of acylcarnitines in livers of 5-week-old $Atg7^{f/f}$;Alb-$Cre$ ($n = 5$) as well as of 12-week-old $Atg5^{f/f}$;Mx1-$Cre$ ($n = 3$) mice, relative to those in the corresponding age-matched control mice, $Atg7^{f/f}$ ($n = 5$) and $Atg5^{f/f}$ ($n = 3$). At the age of 10 weeks, $Atg5^{f/f}$ and $Atg5^{f/f}$;Mx1-$Cre$ mice were intraperitoneally injected with pIpC to delete Atg5 in the liver. **c** Relative levels of carnitines in livers of mice described in **b**. **d**–**f** Blood glucose (**d**), free-fatty acid (**e**), and β-OHB (**f**) in mice described in **b**. **g** Tracer study using [$^{13}$C] palmitate. pIpC-injected $Atg5^{f/f}$ and $Atg5^{f/f}$;Mx1-$Cre$ mice were starved for 24 h and administered [$^{13}$C] palmitate 1 h before sacrifice. Total (left graph) and $^{13}$C-labeled (right graph) β-OHB were analyzed by liquid chromatography–tandem mass spectrometry. The experiments were performed three times. Data are means ± s.e.m. *$P < 0.05$, **$P < 0.01$, and ***$P < 0.001$ as determined by Welch's $t$-test

**Decreased PPARα protein in autophagy-deficient livers**. We speculated that reduced expression of genes encoding enzymes related to lipid oxidation in autophagy-deficient mouse livers was causing these changes. Indeed, the expression level of such genes was markedly suppressed in livers of both *Atg7*f/f;Alb-*Cre* (Fig. 2a) and pIpC-injected *Atg5*f/f;Mx1-*Cre* mice, even under fed conditions (Supplementary Figure 4a). Fasting increased the transcript levels of β-oxidation-related genes such as *Carnitine O-palmitoyltransferase 1, liver isoform* (Cpt1a), *Carnitine O-palmitoyltransferase 2, mitochondrial* (Cpt2), and *carnitine-acylcarnitine translocase* (Cact) in both control and mutant livers, but the level of induction was significantly lower in mutant livers than in controls (Fig. 2a and Supplementary Figure 4a).

In the next series of experiments, we investigated the levels of the nuclear receptor PPARα, a master regulator of lipid metabolism, and its co-repressor NCoR1. Remarkably, PPARα protein was barely detectable in nuclear fractions of livers of *Atg7*f/f;Alb-*Cre* mice, regardless of nutrient conditions (Fig. 2b). The level of *PPARα* mRNA was lower in mutant livers than in control livers (Fig. 2b), consistent with the idea that PPARα regulates its own expression. In marked contrast to the reduction in PPARα, we observed prominent accumulation of NCoR1 in both nuclear and cytoplasmic fractions of livers of *Atg7*f/f;Alb-*Cre* mice, not accompanied by upregulation of *NCoR1* gene expression (Fig. 2a, b). Fasting decreased NCoR1 in both fractions of mutant livers, but the levels were still higher than those in control livers (Fig. 2b). In our experimental setting, we did not detect NCoR1 in both cytosolic and nuclear fractions of liver homogenates of *Atg7*f/f mice (Fig. 2b). But, NCoR1 was clearly recognized in both nuclear and cytoplasmic fractions of primary hepatocytes isolated from *Atg7*f/f mice, and they decreased upon ketogenic conditions (Fig. 2c). Similar to in vivo analysis, the deletion of *Atg7* by expression of Cre-recombinase was accompanied by the accumulation of NCoR1 (Fig. 2c). Expression of PPARα target genes was inversely correlated to NCoR1 level (Fig. 2d). In agreement with the biochemical data, immunohisto-fluorescence analysis revealed higher signal intensities of nuclear NCoR1 in *Atg7*-deficient hepatocytes than in control hepatocytes under both fed and fasting conditions (Fig. 2e).

**Suppression of β-oxidation dependent on NCoR1**. To examine the regulation of PPARα through autophagy in more detail, we deleted *ATG7* in the human hepatocyte carcinoma cell line HepG2, using CRISPR/Cas9 technology. Consistent with previous reports[4,34], conversion of LC3-I to LC3-II was impaired in the knockout cells (Fig. 3a). p62 markedly accumulated in the mutant cells (Fig. 3a), and expression of Nrf2-target genes such as *NAD (P)H quinone dehydrogenase 1* (NQO1), *glutamate-cysteine ligase catalytic subunit* (GCLC), and *UDP-glucose 6-dehydrogenase* (UGDH) was significantly induced (Fig. 3b). Not surprisingly, loss of *ATG7* caused the accumulation of NCoR1 in both cyto-solic and nuclear fractions of HepG2 cells (Fig. 3a), resulting in suppression of gene expression of PPARα target genes (Fig. 3b). Chromatin immunoprecipitation (ChIP) with an antibody against acetylated lysine 27 of histone H3 (H3K27ac), which is an enhancer mark, coupled with quantitative PCR showed that deposition of H3K27ac in the promoter regions of PPARα target genes such as *CPT1A* and *CPT2* was significantly diminished in the absence of *ATG7* (Fig. 3c). Such decrease was not obvious in the case of the promoter of an Nrf2-target, *NQO1* (Fig. 3c). *GATA1* exon 3 was examined as a negative control locus. These results suggest that the increase of NCoR1 recruitment to the regulatory regions of PPARα target genes in *ATG7*-deficient cells suppresses the enhancer formation, which is consistent with the function of NCoR1 as a co-repressor.

Knockdown of *p62* in *ATG7*-deficient HepG2 cells suppressed expression of Nrf2-target genes, but had no effect on the repression of PPARα target genes (Fig. 4a, b). Conversely, suppression of PPARα target genes in *ATG7*−/− HepG2 cells was abolished by simultaneous knockdown of *NCoR1* (Fig. 4c, d). In accordance with a previous study of liver-specific *NCoR1*-knockout mice[35], loss of *NCoR1* in HepG2 cells led to prominent induction of PPARα target genes (Fig. 4e, f). In both wild-type and *NCoR1*-knockout HepG2 cells, expression of PPARα target genes was repressed by overexpression of NCoR1 (Fig. 4e, f). These results suggest that quantitative control of NCoR1 has an impact on expression of PPARα target genes.

In the next series of experiments, we examined the effect of accumulation of p62 and/or NCoR1 in autophagy-incompetent mouse livers. As previously reported[36], severe hepatomegaly and liver injury observed in hepatocyte-specific *Atg7*-knockout mice were ameliorated by concomitant loss of *p62* (Supplementary Figure 5a and b, upper graphs), and induced expression of Nrf2-target genes in *Atg7*f/f;Alb-*Cre* mouse livers was robustly down-regulated (Supplementary Figure 5c, upper graphs). On the other hand, reduced expression of PPARα target genes was still observed in livers of *Atg7*f/f;*p62*f/f;Alb-*Cre* mice, in which both *Atg7* and *p62* were deleted in hepatocytes, regardless of nutrient conditions (Fig. 5a). We confirmed the significant reduction of nuclear PPARα levels, as well as the marked accumulation of both nuclear and cytoplasmic NCoR1, in livers of *Atg7*f/f;*p62*f/f;Alb-*Cre* mice (Fig. 5b). Such sharp fluctuations were detected even in fasting conditions (Fig. 5b). Likewise, the decreased expression of PPARα-targets in the *Atg5*−/− livers was not ameliorated by the additional deletion of *p62* (Supplementary Figure 4b). Next, we crossbred *Atg7*f/f;Alb-*Cre* with *NCoR1*f/f mice[37] to generate hepatocyte-specific *Atg7/NCoR1* double-knockout mice. Conco-mitant loss of NCoR1 in *Atg7*-knockout mouse livers abolished the suppression of PPARα target genes, albeit not completely (Supplementary Figure 6a). As with liver-specific *Atg7*-single knockout mice, *Atg7*f/f;*NCoR1*f/f;Alb-*Cre* mice exhibited severe liver enlargement and injury (Supplementary Figure 5a and b, bottom graphs), raising the possibility that this pathological condition influences expression of PPARα target genes. To exclude this possibility, we generated *Atg7*f/f;*p62*f/f;*NCoR1*f/f;Alb-*Cre* mice. Liver phenotypes in *Atg7*f/f;*NCoR1*f/f;Alb-*Cre* mice were substantially ameliorated by simultaneous ablation of *p62*: both hepatomegaly and leakage of hepatic enzymes into sera, observed in *Atg7*f/f;*NCoR1*f/f;Alb-*Cre* mice, were largely abrogated by additional loss of *p62* (Supplementary Figure 5a and b, bottom graphs). Loss of *p62* also decreased the induction of Nrf2-target genes in livers of *Atg7*f/f;*NCoR1*f/f;Alb-*Cre* mice (Supplementary Figure 5c, bottom graphs). Remarkably, the suppression of PPARα target genes in *Atg7*−/− *p62*−/− double-knockout livers was alleviated by additional loss of *NCoR1* (Fig. 5c). In stark contrast to the significant accumulation of NCoR1 in both the nuclear and cytoplasmic fractions of *Atg7 p62* double-knockout livers (Fig. 5b), the signal was abolished by additional loss of *NCoR1* (Fig. 5d). Consequently, the reduction in the nuclear level of PPARα in *Atg7 p62* double-knockout livers was rescued by simultaneous loss of *NCoR1* (Fig. 5d). Immunohistofluorescence microscopy confirmed that the signal intensity of nuclear NCoR1 was higher in *Atg7*−/− *p62*−/− hepatocytes than in control hepatocytes (Fig. 5e). As expected, no nuclear signals were recognized in *Atg7 p62 NCoR1* triple-knockout hepatocytes, which validated the signals observed in other genotypes (Fig. 5e). Further, the blood level of β-OHB in *Atg7*f/f;*p62*f/f;Alb-*Cre* mice was still lower than in control mice under fasting conditions, which was restored by additional loss of *NCoR1* (Fig. 5f). Although the expression of PPARα target genes was not influenced by single deletion of *p62*, it was significantly induced

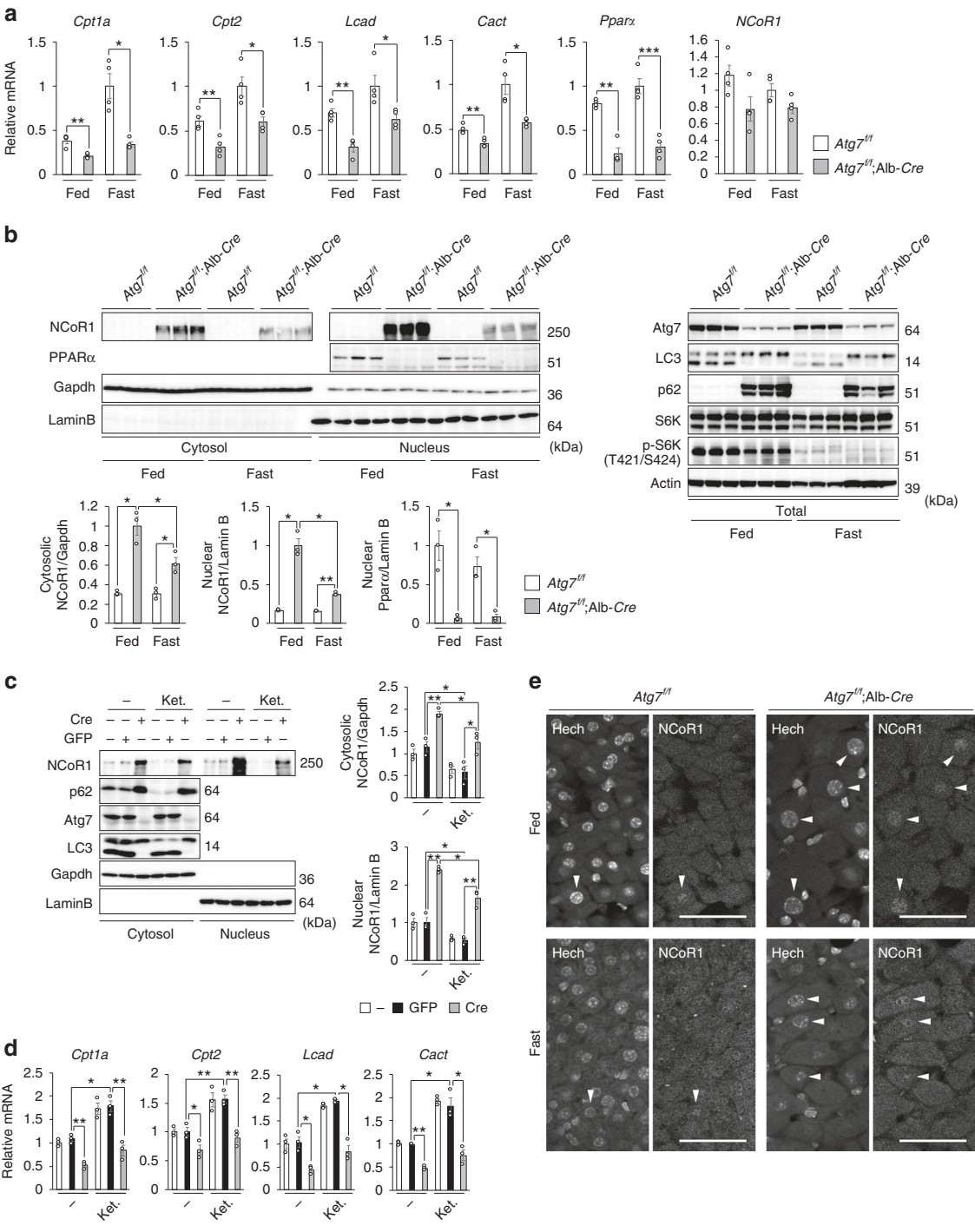

**Fig. 2** Accumulation of NCoR1 and inactivation of PPARα in autophagy-deficient livers. **a** Gene expression of enzymes related to lipid oxidation in *Atg7*-deficient livers. Total RNAs were prepared from livers of *Atg7^f/f* (*n* = 4) and *Atg7^f/f*;Alb-*Cre* (*n* = 4) mice aged at 5 weeks under both fed and fasting conditions. Values were normalized against the amount of mRNA in the livers of fasting *Atg7^f/f* mice. The experiments were performed three times. **b** PPARα and NCoR1 level in *Atg7*-deficient mouse livers. Total homogenate, as well as nuclear and cytoplasmic fractions, were prepared from livers of *Atg7^f/f* and *Atg7^f/f*;Alb-*Cre* mice aged 5 weeks, and subjected to immunoblotting with the indicated antibodies. Bar graphs indicate the quantitative densitometric analyses of indicated cytoplasmic and nuclear proteins relative to Gapdh and Lamin B, respectively. **c** Immunoblot analysis. Primary hepatocytes were isolated from *Atg7^f/f* mice and infected with adenovirus for GFP or Cre-recombinase. Forty-eight hours after infection, the cells were cultured under nutrient-rich and ketogenic conditions for 24 h, and then both nuclear and cytoplasmic fractions were prepared from the cells and subjected to immunoblotting with the indicated antibodies. Data are representative of three separate experiments. Bar graphs indicate the quantitative densitometric analyses of cytoplasmic and nuclear NCoR1 relative to Gapdh and Lamin B, respectively. **d** Total RNAs were prepared from cells described in **c**. Values were normalized against the amount of mRNA in adenovirus-uninfected *Atg7^f/f* primary hepatocytes. The experiments were performed three times. **e** Immunohistofluorescence microscopy. Liver sections of *Atg7^f/f* and *Atg7^f/f*;Alb-*Cre* mice aged 5 weeks were immunostained with anti-NCoR1 antibody. Arrowheads indicate intranuclear signals for NCoR1. Bars: 50 μm. Data are means ± s.e.m. *$P < 0.05$, **$P < 0.01$, and ***$P < 0.001$ as determined by Welch's *t*-test

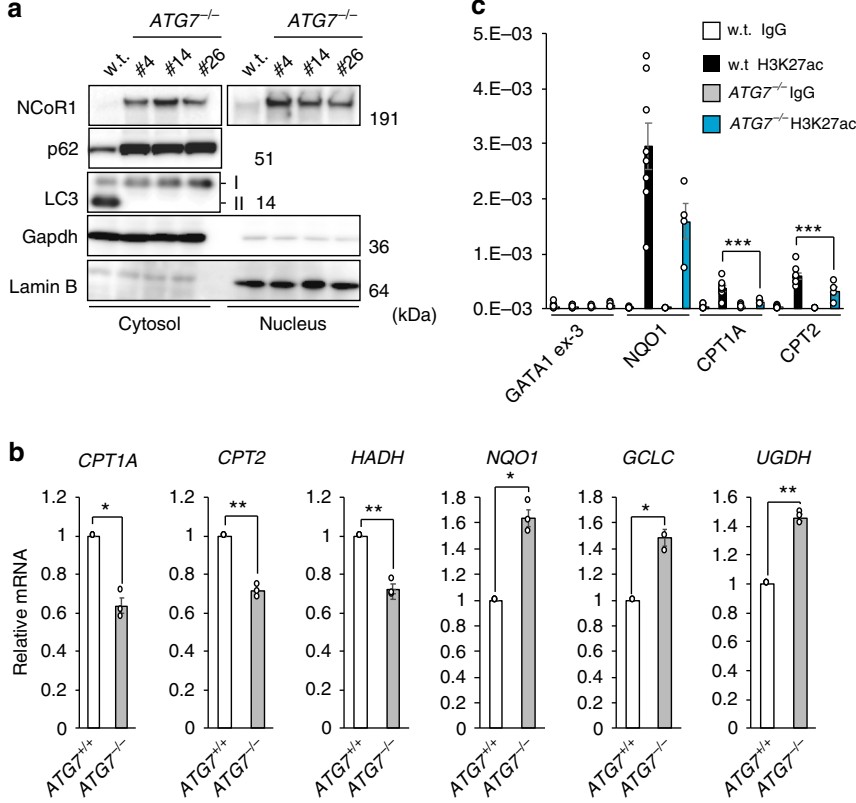

**Fig. 3** Decreased H3K27ac deposition on promoter regions of PPARα targets in autophagy-incompetent cells. **a** Immunoblot analysis. Both nuclear and cytoplasmic fractions were prepared from three independent ATG7-knockout HepG2 cells and subjected to immunoblotting with the indicated antibodies. Data are representative of three separate experiments. **b** Real-time PCR analysis. Total RNAs were prepared from parental HepG2 and ATG7-deficient HepG2 (#14) cells. Values were normalized against the amount of mRNA in parental HepG2 cells. The experiments were performed three times. **c** H3K27ac deposition was compared between wild-type and ATG7- knockout HepG2 cells (#4) by ChIP assay at indicated genomic regions. GATA1-ex3 and NQO1 promoter regions were used as negative and positive controls, respectively. Data are average ± s.e.m. from four independent experiments. Statistical analysis was performed by Welch's $t$-test. $*P < 0.05$, and $**P < 0.01$

in NCoR1-knockout mouse livers (Supplementary Figure 6b and c). On the basis of those in vitro and in vivo results, we concluded that downregulation of lipid-oxidation in autophagy-deficient livers is mainly due to accumulation of NCoR1.

**NCoR1 is a substrate for autophagy.** How does autophagy affect the level of NCoR1? First, we examined the kinetics of NCoR1 in response to starvation under autophagy-intact or -defective conditions. Amino-acid starvation caused a marked decrease in NCoR1 levels in both nuclear and cytoplasmic fractions of control siRNA-treated HepG2 cells (Fig. 6a). This reduction was largely inhibited by knockdown of ATG7 (Fig. 6a). The treatment of wild-type HepG2 cells with the lysosomal protease inhibitors E64d and pepstatin A caused a marked increase in the NCoR1 level (Fig. 6b). The amounts of both nuclear and cytoplasmic NCoR1 in $ATG7^{-/-}$ HepG2 cells decreased upon expression of wild-type ATG7, but not expression of GFP or an active-site mutant of ATG7, ATG7$^{C572S}$ (Fig. 6c). Starvation promoted the ATG7-mediated decrease in the NCoR1 level (Fig. 6c). Oxygen consumption rate (OCR) in ATG7-deficient HepG2 cells under β-oxidation conditions tended to be lower than in cells expressing wild-type ATG7. Moreover, the addition of etomoxir, which binds irreversibly to the CPT1A transporter and inhibits fatty-acid oxidation, barely had an effect on the OCR (Fig. 6d). In marked contrast, the OCR in $ATG7^{-/-}$ HepG2 cells expressing ATG7 significantly decreased in response to treatment with etomoxir (Fig. 6d).

Immunofluorescence staining revealed that endogenous NCoR1 and an autophagosome-localizing protein, GABARAP (gamma-aminobutyric acid receptor-associated protein), co-localized in cytoplasmic puncta structures of HepG2 cells (Fig. 7a), although we could not investigate the co-localization of NCoR1 with LC3 due to the lack of a monoclonal antibody for LC3. Most (55.4 ± 2.98%) of cytoplasmic NCoR1 punctae were positive for GABARAP. Consistent with the biochemical results (Fig. 6a), amino-acid starvation for 12 h decreased the number of such structures, probably due to autophagic degradation (Fig. 7a). The treatment of HepG2 with bafilomycin A₁, an inhibitor of vacuolar H⁺-ATPases, increased the structures positive for both GABARAP and NCoR1 (Supplementary Figure 7). We also carried out immunofluorescent analysis with anti-LAMP1 and anti-NCoR1 antibodies. As expected, both proteins were co-localized in wild-type but not Atg7-deficient HepG2 cells (Fig. 7b). In ATG7-knockout HepG2 cells, NCoR1 formed small punctate structures in both nucleus and cytoplasm, in addition to the large cytoplasmic puncta (Fig. 7a). Among these structures, GABARAP mainly co-localized to large cytosolic NCoR1-positive structures, regardless of nutrient conditions (Fig. 7a). Unlike wild-type HepG2 cells, the abundance of these structures did not decrease upon starvation (Fig. 7a). Our previous genetic studies showed the formation of LC3-dots through the interaction of LC3 with p62 even in Atg7-deficient mouse hepatocytes[4,36]. Likewise, excessive accumulation of NCoR1 might sequester GABARAP due to their physical interaction, irrespective of the lipidation of GABARAP.

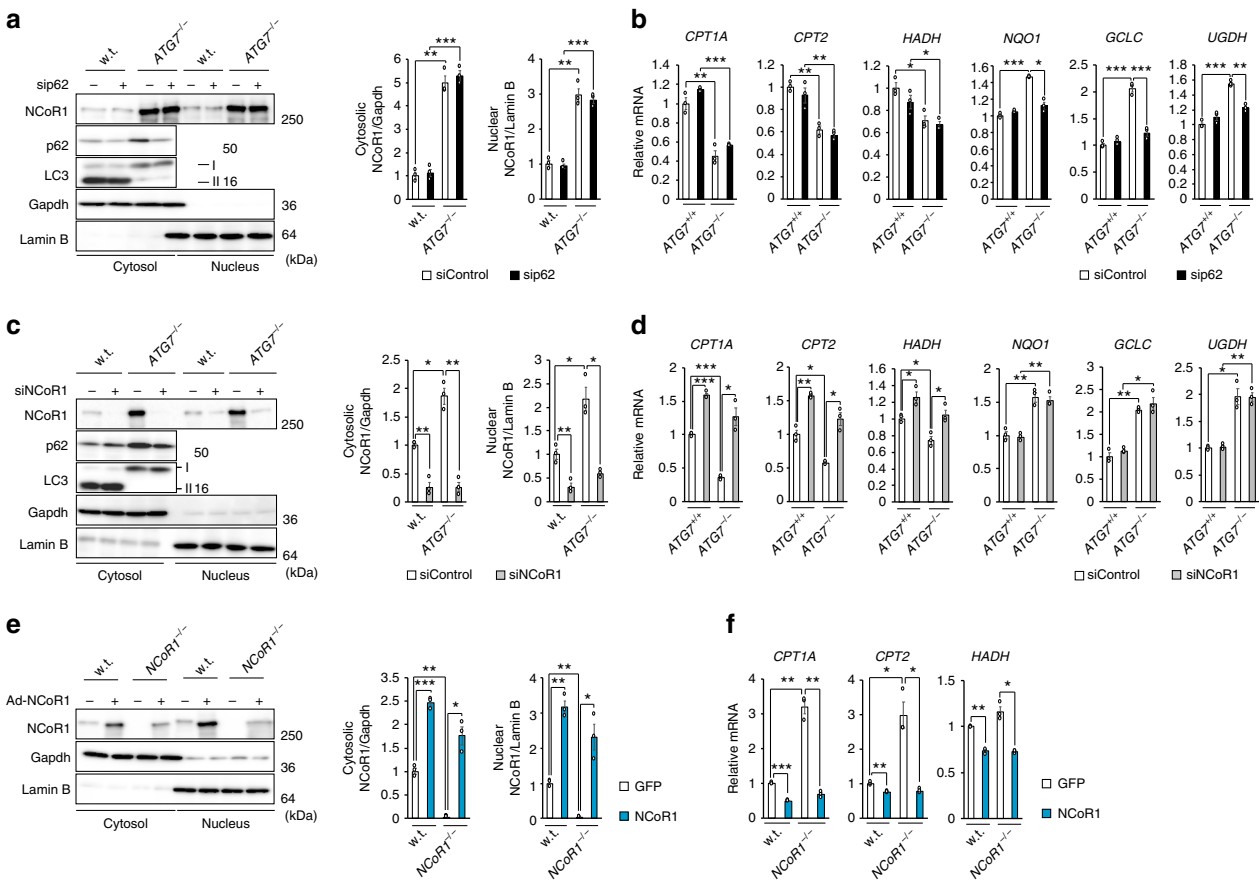

**Fig. 4** NCoR1-dependent PPARα-inactivation in autophagy-incompetent cells. **a**, **c** Immunoblot analysis. Parental and *ATG7*-knockout HepG2 cells (#14) were treated with siRNA for *p62* (**a**) and *NCoR1* (**c**). Thereafter, both nuclear and cytoplasmic fractions were prepared and subjected to immunoblotting with the indicated antibodies. Data are representative of three separate experiments. Bar graphs indicate the quantitative densitometric analyses of cytoplasmic and nuclear NCoR1 relative to Gapdh and Lamin B, respectively. **b**, **d** Real-time PCR analysis. Total RNAs were prepared from cells described in **b** and **d**. Values were normalized against the amount of mRNA in parental HepG2 cells treated with control siRNA. The experiments were performed three times. **e** Immunoblot analysis. GFP or NCoR1 was exogenously expressed in wild-type and *NCoR1*-knockout HepG2 cells using the adenovirus system. Forty-eight hours after infection, both nuclear and cytoplasmic fractions were prepared from cells and subjected to immunoblotting with the indicated antibodies. Data shown are representative of three separate experiments. Bar graphs indicate the quantitative densitometric analyses of cytoplasmic and nuclear NCoR1 relative to Gapdh and Lamin B, respectively. **f** Real-time PCR analysis. Total RNAs were prepared from cells described in **e**. Values were normalized against the amount of mRNA in GFP-expressing wild-type HepG2 cells. The experiments were performed three times. Data are means ± s.e.m. *P < 0.05, **P < 0.01, and ***P < 0.001 as determined by Welch's *t*-test

Next, to investigate whether NCoR1 interacts with LC3 and/or GABARAP, we expressed One-Strep-FLAG-tagged LC3B or GABARAP family proteins (GABARAP, gamma-aminobutyric acid receptor-associated protein-like [GABARAPL]1 and GABARAPL2) and performed pull-down assays. HEK293T cells were utilized for the pull-down assays due to their high transfection efficiency and protein production. In agreement with previous reports[38], these assays confirmed the specific binding of ULK1 to GABARAP family proteins, as well as binding of p62 to both LC3B and GABARAP family proteins (Fig. 8a), indicating that our experimental setting works as expected. Similar to ULK1, NCoR1 bound to GABARAP family proteins, but not LC3B (Fig. 8a). To determine which domain of NCoR1 is required for its interaction with GABARAP family proteins, we constructed a series of NCoR1-deletion mutants (Fig. 8b, left panel) and performed pull-down assays with the One-Strep-FLAG-tagged GABARAP. NCoR1 N (aa 1–674) and NCoR1 ΔC (aa 1–1149) were clearly detected in the precipitant (Fig. 8b, middle panel). However, deletion mutants NCoR1 M (aa 675–1500), NCoR1 C (aa 1501–2440), and NCoR1 ΔN (1150–2440) exhibited a marked decrease in binding to

GABARAP (Fig. 8b, middle panel). To narrow down the interaction domain, we prepared a series of deletions starting from NCoR1 ΔC (N1–N4) (Fig. 8b, left panel) and performed pull-down assays. These assays revealed that NCoR1 N1, covering amino acids 1–370, is sufficient for the interaction between NCoR1 and GABARAP (Fig. 8b, right panel). We noticed that a region within NCoR1 N1 (aa 344–349) contained a potential LC3-interacting region (LIR)[27]/GABARAP-interacting motif (GIM)[39], KQFPEI, which is conserved in species as divergent as *C. elegans* (Fig. 8c). Deletion of the FPEI sequence (NCoR1 ΔLIR/GIM) limited the ability of NCoR1 to bind GABARAP, as judged by pull-down assay (Fig. 8d).

To investigate whether NCoR1 is degraded by autophagy in a LIR/GIM-dependent manner, we expressed FLAG-tagged wild-type and ΔLIR/GIM NCoR1 in *NCoR1*-deficient HepG2 cells. Both wild-type and mutant were efficiently expressed in the cells (Fig. 9a). Similar to the dynamics of endogenous NCoR1 (Fig. 6a), exogenous wild-type NCoR1 decreased in response to amino-acid starvation, which was not the case for NCoR1 ΔLIR/GIM (Fig. 9a). Consistent with this biochemical data, immunofluorescence analysis showed that FLAG-NCoR1 but not NCoR1

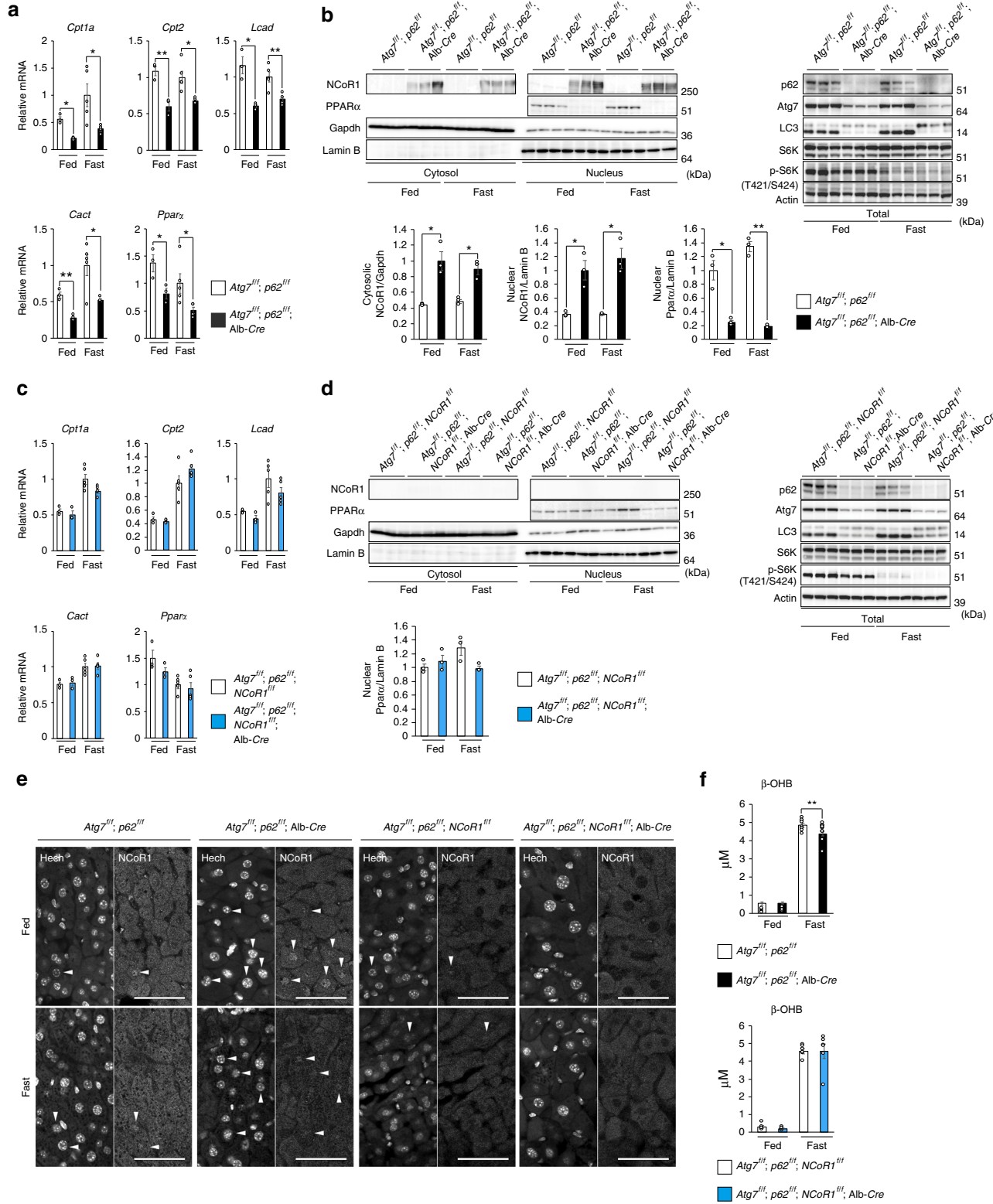

ΔLIR/GIM was delivered to lysosome (Fig. 9b). Furthermore, GIM-deleted NCoR1 has a higher suppressive effect on the expression of PPARα target genes than wild-type NCoR1 under amino-acid starvation conditions (Fig. 9c). Taken together, these results strongly suggest that NCoR1 is selectively degraded by autophagy in a LIR/GIM-dependent manner and that it is indispensable for transactivation of PPARα in response to starvation.

## Discussion

NCoR1 serves as scaffold that facilitates interaction of several docking proteins to fine-tune transactivation of transcription factors, in particular nuclear receptors that have important roles in metabolic control. Interaction of NCoR1 with nuclear receptors and histone deacetylases is vital for nuclear receptor-mediated downregulation of gene expression[40]. Besides the dissociation between NCoR1 and nuclear receptors

**Fig. 5** NCoR1-dependent PPARα-inactivation in autophagy-deficient livers. **a, c** Gene expression of enzymes related to lipid oxidation in *Atg7 p62* double- (**a**) and *Atg7 p62 NCoR1* triple-knockout livers (**c**). Total RNAs were prepared from livers of *Atg7[f/f];p62[f/f]* (fed: n = 3; fasted: n = 5), *Atg7[f/f];p62[f/f]*;Alb-*Cre* (fed: n = 3; fasted: n = 4), *Atg7[f/f];p62[f/f];NCoR1[f/f]* (fed: n = 4; fasted: n = 4), and *Atg7[f/f];p62[f/f];NCoR1[f/f]*;Alb-*Cre* (fed: n = 4; fasted: n = 4) mice aged 5 weeks under both fed and fasting conditions. Values were normalized against the amount of mRNA in the livers of fasting *Atg7[f/f];p62[f/f]* or *Atg7[f/f];p62[f/f];NCoR1[f/f]* mice. The experiments were performed three times. **b, d** NCoR1 level in *Atg7 p62* double- (**b**) and *Atg7 p62 NCoR1* triple-knockout livers (**d**). Total homogenate, as well as nuclear and cytoplasmic fractions, were prepared from livers of mice of the indicated genotypes and subjected to immunoblotting with the indicated antibodies. Bar graphs indicate the quantitative densitometric analyses of indicated cytoplasmic and nuclear proteins relative to Gapdh and Lamin B, respectively. **e** Immunohistofluorescence microscopy. Liver sections of *Atg7[f/f];p62[f/f]*, *Atg7[f/f];p62[f/f]*;Alb-*Cre*, *Atg7[f/f];p62[f/f];NCoR1[f/f]*, and *Atg7[f/f]; p62[f/f];NCoR1[f/f]*;Alb-*Cre* mice aged 5 weeks were immunostained with anti-NCoR1 antibody. Arrowheads indicate intranuclear signals for NCoR1. Bars: 50 μm. **f** Blood β-OHB in mice described in **a**. Data are means ± s.e.m. *P < 0.05, **P < 0.01, and ***P < 0.001 as determined by Welch's t-test

in response to ligands/agonists, this repression is modulated at multiple levels, including gene expression, post-translational modifications such as ubiquitination and phosphorylation, and proteasomal degradation of NCoR1[41]. In this study, we found that autophagy also participates in regulation of the activity of the nuclear receptor PPARα through degradation of NCoR1, and that suppression of liver autophagy is accompanied by defective β-oxidation and ketogenesis. Under nutrient-rich conditions, mechanistic target of rapamycin complex 1 (mTORC1) phosphorylates ribosomal protein S6 kinase 2 (S6K2). NCoR1 forms a complex with the S6K2, and the resultant complex translocates into the nucleus to suppress genes encoding enzymes involved in β-oxidation[42]. mTORC1 also binds and phosphorylates TFEB, a master transcription factor for a battery of *Atg* and lysosomal genes, causing it to be retained in the cytoplasm[43–45]. In response to nutrient deprivation and subsequent mTORC1 inactivation, nuclear translocation of NCoR1 is inhibited, TFEB is concomitantly dephosphorylated, and ULK1 kinase, an upstream factor involved in autophagosome formation, is activated. Consequently, both autophagy and β-oxidation followed by production of ketone body occur at the same time[46]. At this time, the autophagic degradation of NCoR1 contributes to PPARα-activation to effectively promote β-oxidation in response to physiological fasting (Fig. 9d). Previously, autophagy was thought to contribute to lipid oxidation by increasing the supply of free fatty acids (lipophagy). However, here we propose that autophagy is integrated into a highly sophisticated regulatory mechanism for a nuclear factor, PPARα, and that for this reason, impairment of autophagy in the liver causes defects in β-oxidation and ketogenesis.

We observe a marked reduction in PPARα in autophagy-deficient mouse livers. However, the phenotypes of liver-specific *Atg7*-knockout mice are likely to be different from those of liver-specific *PPARα*-knockout mice, which have higher levels of liver triglyceride and cholesterol ester during fasting and exhibit severe steatosis[47]. In fact, we recognized a slight increase in the levels of some molecular species of triglyceride and cholesterol esters in *Atg7*-deficient mouse livers (Supplementary Figure 1). Repression of the transactivation of nuclear receptors by NCoR1 in livers is not restricted to PPARα. For example, NCoR1 modulates transactivation of nuclear receptors such as LXRα and estrogen-related receptor α (ERRα) other than PPARα[41]. Thus, suppression of liver autophagy should be accompanied by repression of multiple nuclear receptors. Indeed, LXRα target genes that encode enzymes involved in lipogenesis are downregulated in mouse livers lacking Rb1cc1 (also called Fip200), a component of the ULK1 kinase complex[21]. Consistent with this, our microarray analysis of *Atg7[−/−]* mouse livers (GEO accession number: GSE65174)[48] revealed a reduced expression of LXRα and PPARα target genes. We verified the NCoR1-dependent suppression of LXRα-targets (Supplementary Figure 8a–d) and of LXRα protein (Supplementary Figure 8e–h) in autophagy-deficient livers. Given

the concurrent suppression of genes related to both lipogenesis and lipid oxidation upon loss of Atg7, lipid metabolism in the mutant livers is apparently stable under normal conditions, but should be stagnant under conditions in which fatty acids are mobilized. Indeed, liver steatosis under physiological fasting and high-fat diet conditions was abolished by loss of *Rb1cc1*, *Atg7*, or *Atg5*[19–21,23]. We also verified the impairment of physiological hepatosteatosis upon fasting in *Atg7[f/f]*;Alb-*Cre* mice (Supplementary Figure 9). In sharp contrast, loss of *NCoR1* in mouse livers causes induction of gene sets regulated by LXRα, PPARα, and ERRα, leading to concurrent induction of lipogenesis and lipid oxidation[35]. In *Atg7 p62 NCoR1* triple-knockout livers, the expression of PPARα target genes recovered up to control levels (Fig. 5a). In contrast to *NCoR1*-knockout livers, however, we did not observe higher induction of the targets (Supplementary Figure 4), implying the presence of a still-hidden suppressive mechanism.

Recently, Sinha et al.[49] reported that phosphorylation of RPS6KB1 induced by knockdown of *ULK1* promotes nuclear translocation of NCoR1. We examined the phosphorylation level of RPS6KB1 in *Atg7*-deficient hepatocytes and found that the level was comparable to that in control cells (Supplementary Figure 10), implying that the nuclear accumulation of NCoR1 in *Atg7*-knockout livers depends on a distinct mechanism from the RPS6KB1-mediated translocation of NCoR1. In addition, Sinha et al. showed that the nuclear accumulation of NCoR1 down-regulates the expression of the *SCD1* gene encoding stearoyl-CoA desaturase, which converts saturated fatty acids (SFAs) to mono-unsaturated fatty acids (MUFAs). As a result, it increases the ration of SFAs/MUFAs and then suppresses lipid droplet formation[49]. *SCD1* is a target of LXRα, and the ration of SFAs/MUFAs in *Atg7*-deficient mouse livers (1.55 ± 0.21/25.2 ± 3.28, ± s.e.m., n = 3) was higher than that in control mouse livers (1.12 ± 0.37/30.23 ± 6.88, ± s.e.m., n = 3). Thus, the increased saturated fatty acids may also affect physiological hepatosteatosis in response to fasting in *Atg7[f/f]*;Alb-*Cre* mice.

In general, selective substrates for autophagy are tagged with a molecular marker that includes ubiquitin, leading to assembly of receptor proteins that bind to both marker molecules and the ATG8 family proteins (including LC3A, LC3B, LC3C, GABARAP, GABARAPL1, and GABARAPL2) near the cargos[50,51]. Most receptor proteins have LIR/GIM domains that mediate their interactions with ATG8 family proteins[27,39]. Ubiquitin-chain and/or receptor proteins on the cargos recruit core autophagy proteins such as FIP200, ULK1, and WIPI1[52,53], beginning the process of autophagosome formation around the targets. Thus, cargo labeling and the transfer of receptor proteins to cargos mainly regulate selective autophagy. In the case of selective autophagy for NCoR1, ubiquitination may serve as a signaling tag. NCoR1 is ubiquitinated by the ubiquitin ligase TBLR1 and subsequently degraded by the proteasome[54,55]. Because NCoR1 translocates from the nucleus to cytoplasm in response to nutrient starvation, ubiquitination of NCoR1 may be

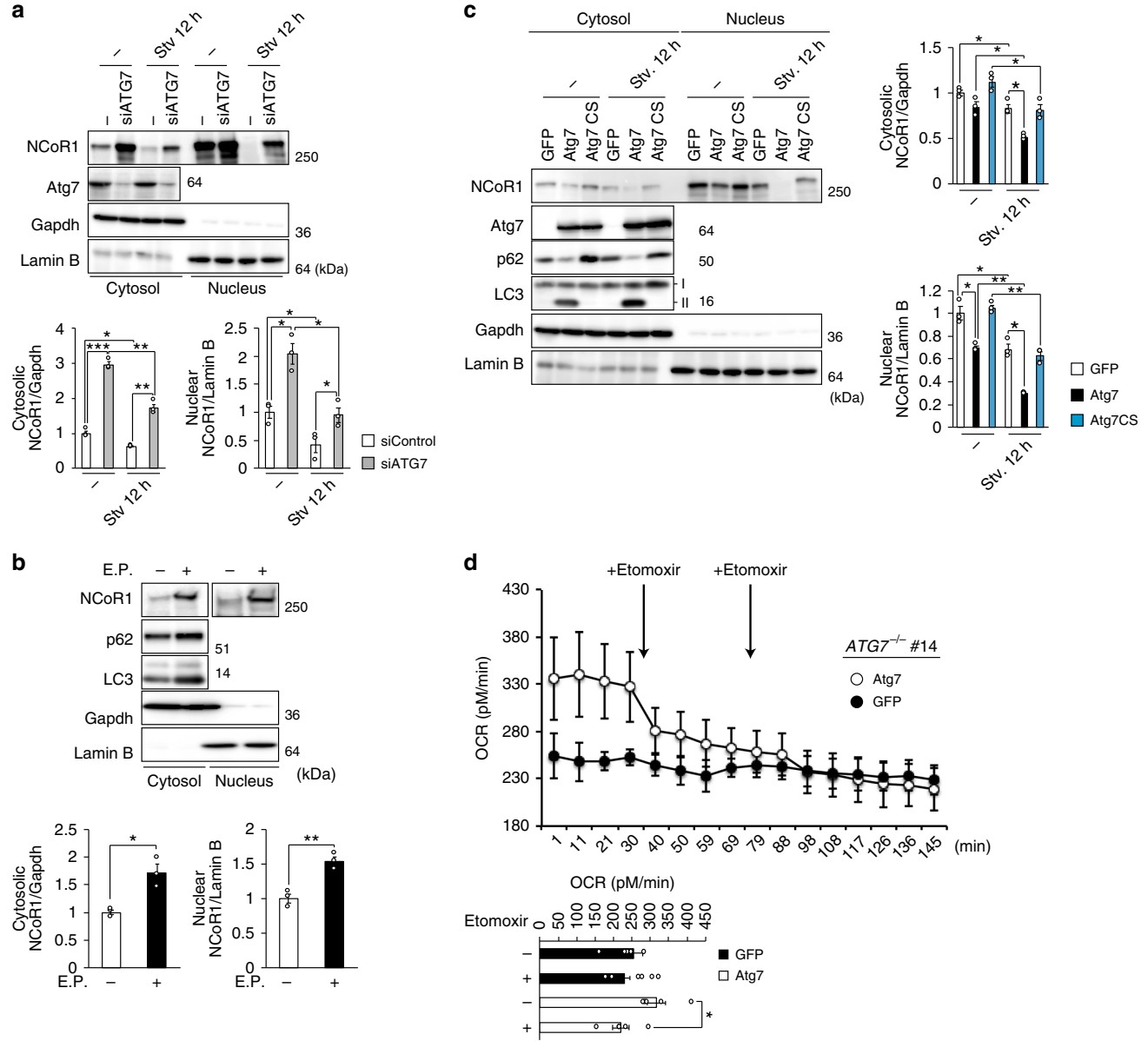

**Fig. 6** Degradation of NCoR1 in autophagy-lysosomal pathway. **a** Immunoblot analysis. Both nuclear and cytoplasmic fractions were prepared from *ATG7*-knockdown HepG2 cells under nutrient-rich and deprived conditions and subjected to immunoblotting with the indicated antibodies. Data are representative of three separate experiments. Bar graphs indicate the quantitative densitometric analyses of cytoplasmic and nuclear NCoR1 relative to Gapdh and Lamin B, respectively. **b** Immunoblot analysis. HepG2 cells were cultured in the presence or absence of E64d and Pepstatin A (E.P.) for 24 h. Subsequently, both nuclear and cytoplasmic fractions were prepared from the HepG2 cells and subjected to immunoblotting with the indicated antibodies. Data are representative of three separate experiments. Bar graphs indicate the quantitative densitometric analyses of cytoplasmic and nuclear NCoR1 relative to Gapdh and Lamin B, respectively. **c** Immunoblot analysis. GFP, wild-type ATG7, or ATG7^C572S was expressed in *ATG7*-knockout HepG2 (#14) cells by adenovirus system. Forty-eight hours after infection, the cells were cultured under nutrient-rich or -deprived conditions. Thereafter, both nuclear and cytoplasmic fractions were prepared and subjected to immunoblotting with the indicated antibodies. Data shown are representative of three separate experiments. Bar graphs indicate the quantitative densitometric analyses of cytoplasmic and nuclear NCoR1 relative to Gapdh and Lamin B, respectively. **d** Oxygen consumption rate (OCR). OCR of *ATG7*^−/− HepG2 cells (#14) expressing GFP or wild-type ATG7 in β-oxidation assay medium was measured using a Seahorse XF24 Extracellular Flux Analyzer. Etomoxir (final concentration 40 μM) was added to the cells after baseline measurement in assay medium. Arrows indicate the time when etomoxir was added to the cells. The graphs represent the average OCR at four time points. Data are means ± s.e.m. *$P < 0.05$, **$P < 0.01$, and ***$P < 0.001$ as determined by Welch's *t*-test

a signal not only for proteasomal degradation but also for selective autophagy, both of which favor the exchange of co-repressors for co-activators. Because fasting decreased the levels of both nuclear and cytoplasmic NCoR1 even in *Atg7*-knockout livers (Fig. 2b), the ubiquitin–proteasome pathway may contribute to the degradation of NCoR1. Notably, although NCoR1 is a selective substrate for autophagy, like receptor proteins for

selective autophagy, it has a LIR/GIM and directly interacts with GABARAP family (Fig. 8). Therefore, NCoR1 is a hybrid protein with characteristics of both autophagy substrates and receptors; accordingly, receptor protein(s) might be dispensable for this type of selective autophagy.

Because both NCoR1 and GABARAP family are ubiquitously expressed, it is plausible that selective autophagy of NCoR1

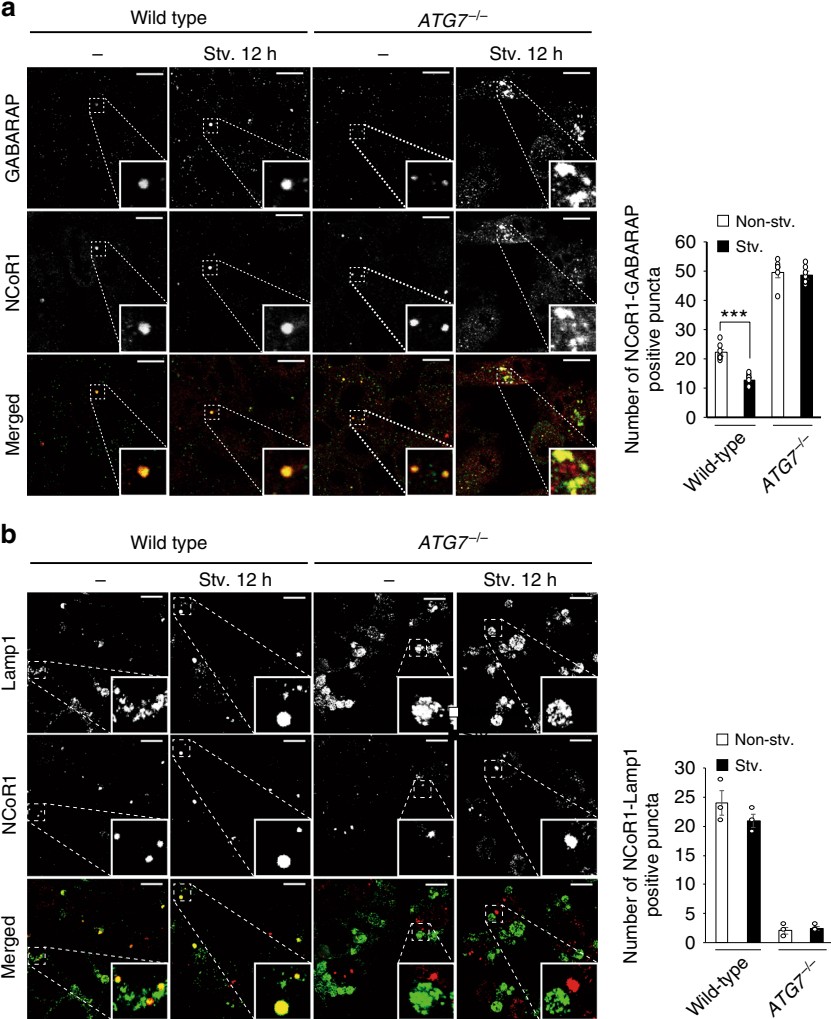

**Fig. 7** Localization of NCoR1 on GABARAP-positive structures. **a, b** Immunofluorescence analysis. Wild-type or *ATG7*-deficient HepG2 cells were cultured under nutrient-rich or -poor conditions, and then immunostained with anti-GABARAP and anti-NCoR1 (**a**) or anti-Lamp1 and anti-NCoR1 antibodies (**b**). The number of cytoplasmic NCoR1 and GABARAP or of cytoplasmic NCoR1 and Lamp1 double-positive dots per 20 cells was counted. The experiments were performed three times. Data are means ± s.e.m. ***$P < 0.001$ as determined by Welch's *t*-test. Each inset is a magnified image. Bar: 2.5 μm

occurs in most tissues. However, the corresponding nuclear receptors for NCoR1 differ according to tissue type, implying that accumulation of NCoR1 due to loss of autophagy has distinct effects among tissues. Suppression of autophagy in metabolic tissues is associated with degeneration and defective differentiation[56]. Although the former is thought to be due to impairment of cellular homeostasis and to p62-mediated Nrf2 activation (in the case of liver), the reason for the latter remains unclear. Recent work with tissue-specific *NCoR1*-knockout mice revealed that loss of NCoR1 results in the activation of distinct transcription factors in specific tissues. For example, expression of PPARγ target genes is specifically induced in adipocyte-specific *NCoR1*-knockout mice, which exhibit increased insulin sensitivity in liver, fat, and muscle, and develop obesity and expansion of fat tissue on a high-fat diet due to an increase in the number of small adipocytes and reduced inflammation in adipose tissue[57]. In muscle-specific *NCoR1*-deficient mice, PPARβ/δ, ERRs, and myocyte-specific enhancer factor 2 (MEF2) are selectively activated, and the mutant mice exhibit increased mitochondrial size and number, increased endurance with more muscle mass and oxidative fibers, and induction of oxidative metabolism[37]. By contrast, adipogenesis is impaired in adipocyte-specific *Atg7* or *Atg5*-knockout

mice, leading to a lean phenotype[58,59]. Loss of *Atg5* or *Atg7* in mouse skeletal muscle causes muscle atrophy and weakness, as well as accumulation of degenerated mitochondria[60,61]. These phenotypes could be explained by the accumulation of NCoR1 and subsequent deregulation of transcription networks, in addition to defective cellular homeostasis.

## Methods

**Cell culture**. HepG2 (ATCC HB-8065) and HEK293T (ATCC CRL-3216) cells were grown in Dulbecco's modified Eagle medium (DMEM) containing 10% fetal bovine serum (FBS), 5 U/mL penicillin, and 50 μg/mL streptomycin. To generate *ATG7*-knockout or *NCoR1*-knockout HepG2 cells, *ATG7* or *NCoR1* guide RNA designed using the CRISPR Design tool (http://crispr.mit.edu/) was subcloned into pX330-U6-Chimeric_BB-CBh-hSpCas9 (Addgene #42230), a human codon-optimized SpCas9 and chimeric guide RNA expression plasmid. HepG2 cells were co-transfected with vectors pX330 and pEGFP-C1 (#6084-1, Clontech Laboratories, Mountain View, CA, USA), and cultured for 2 days. Thereafter, GFP-positive cells were sorted and expanded. Loss of *ATG7* or *NCoR1* was confirmed by heteroduplex mobility assay followed by immunoblot analysis with anti-ATG7 or anti-NCoR1 antibody. For knockdown experiments, HepG2 cells were transfected with 25 nM SMARTpool siRNA for *ATG7*, *p62*, or *NCoR1* using Dharmafect 1 (Thermo Fisher Scientific, Waltham, MA, USA). NCoR1 was expressed using a helper-dependent adenovirus vector system[62] containing loxP at position 143[63]. To exogenously express GFP, ATG7, or ATG7^C572S, we used the Adenovirus Expression Vector Kit (Takara Bio, Kusatsu, Japan). Cells were plated onto 6-well

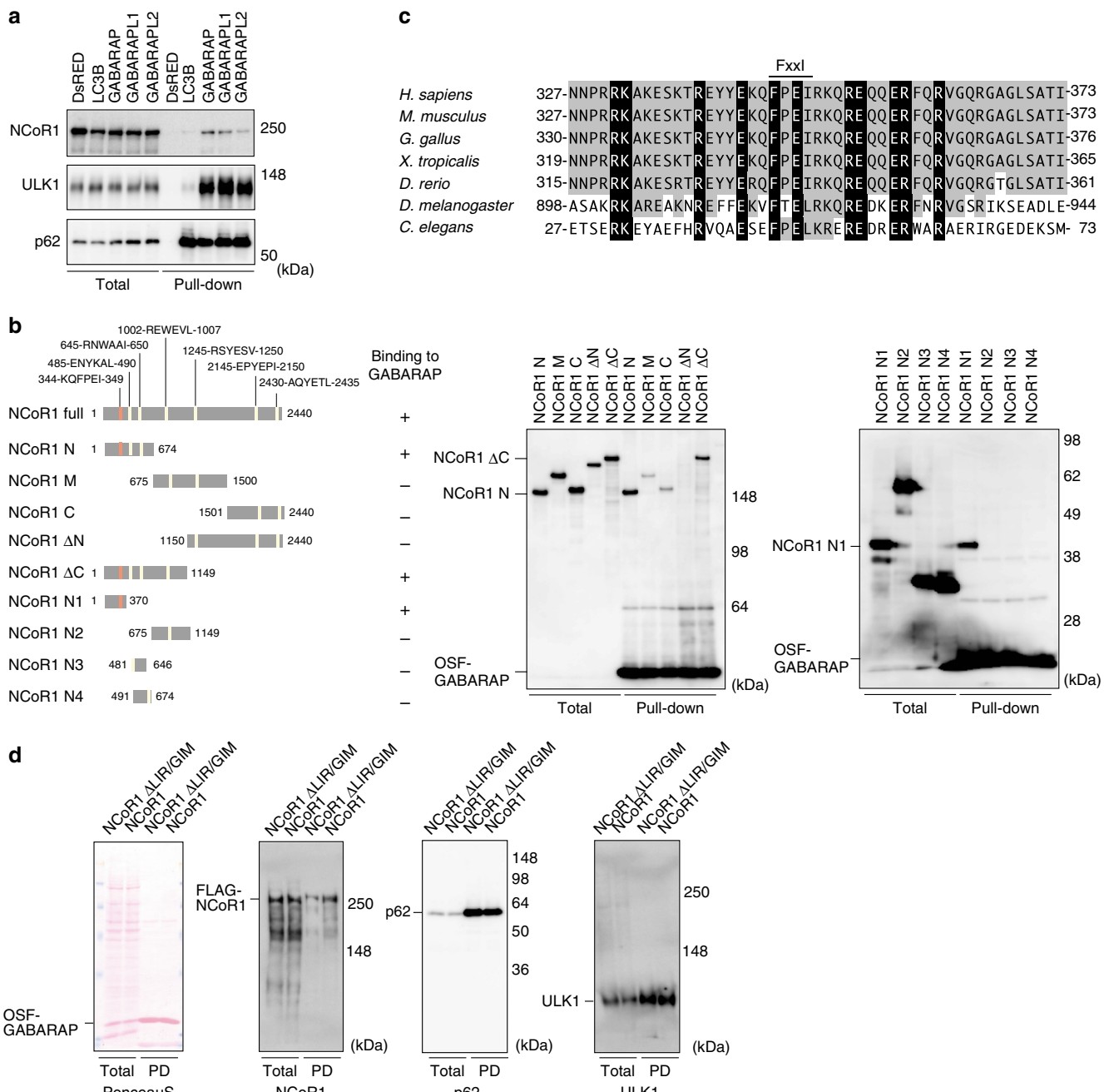

**Fig. 8** Specific interaction of NCoR1 with GABARAP. **a** Pull-down assay. One-Strep-FLAG (OSF)-tagged LC3B or GABARAP family proteins were expressed in HEK293T cells. Precipitates generated with Strep-Tactin Sepharose were subjected to immunoblot analysis with indicated antibodies. ULK1 and p62 bind to GABARAP family and all Atg8 family proteins, respectively, and were therefore used as positive controls. Data are representative of three independent experiments. **b** Diagrams of the deletion-mutation constructs of NCoR1 (left panel) and the corresponding pull-down assays (middle and right panels). Middle panel: Each FLAG-tagged NCoR1-deletion mutant and OSF-GABARAP was co-expressed in HEK293T cells. At 48 h post-transfection, lysates were prepared and precipitated with Strep-Tactin Sepharose. Right panel: OSF-GABARAP immobilized on Strep-Tactin Sepharose was mixed with lysates prepared from cells expressing each NCoR1-deletion mutant. The resultant precipitates were subjected to immunoblotting with anti-FLAG antibody. Data are representative of three independent experiments. **c** Alignment of the LC3-interacting region (LIR)/GABARAP-interacting motif (GIM) of NCoR1 homologs in various species. Black and gray boxes indicate identical amino-acid residues with complete and partial conservation, respectively. **d** Pull-down assay. OSF-GABARAP and FLAG-NCoR1 or NCoR1 ΔLIR/GIM were co-expressed in HEK293T cells. The assay was carried out as described in **a**

dishes in 2 mL of growth medium 24 h before infection. The medium was replaced with fresh medium containing adenovirus with a multiplicity of infection (MOI) of 50. Cells were lysed 48 h later, and lysates were analyzed by immunoblotting. To inhibit degradation in lysosomes, HepG2 cells were treated with E64d (10 μg/mL) and pepstatin A (10 μg/mL), and HepG2 cells were treated with Bafilomycin A$_1$ (0.01 μM final, Wako, 023-11641). E64d and Pepstatin A were from Peptide Institute (Osaka, Japan).

**Measurement of cellular oxygen consumption rate**. HepG2 cells were spread onto 10-cm dishes and cultured in DMEM containing 10% FBS. Twelve hours after plating, the medium was replaced with fresh medium containing adenovirus vector. After 24 h, infected HepG2 cells were seeded on XF24 cell culture plates (50,000 cells per well) and cultured in 500 μL DMEM containing 10% FBS overnight. The culture medium was removed from each well and replaced with 375 μL fatty-acid oxidation (FAO) medium consisting of 111 mM NaCl, 4.7 mM KCl, 1.25 mM

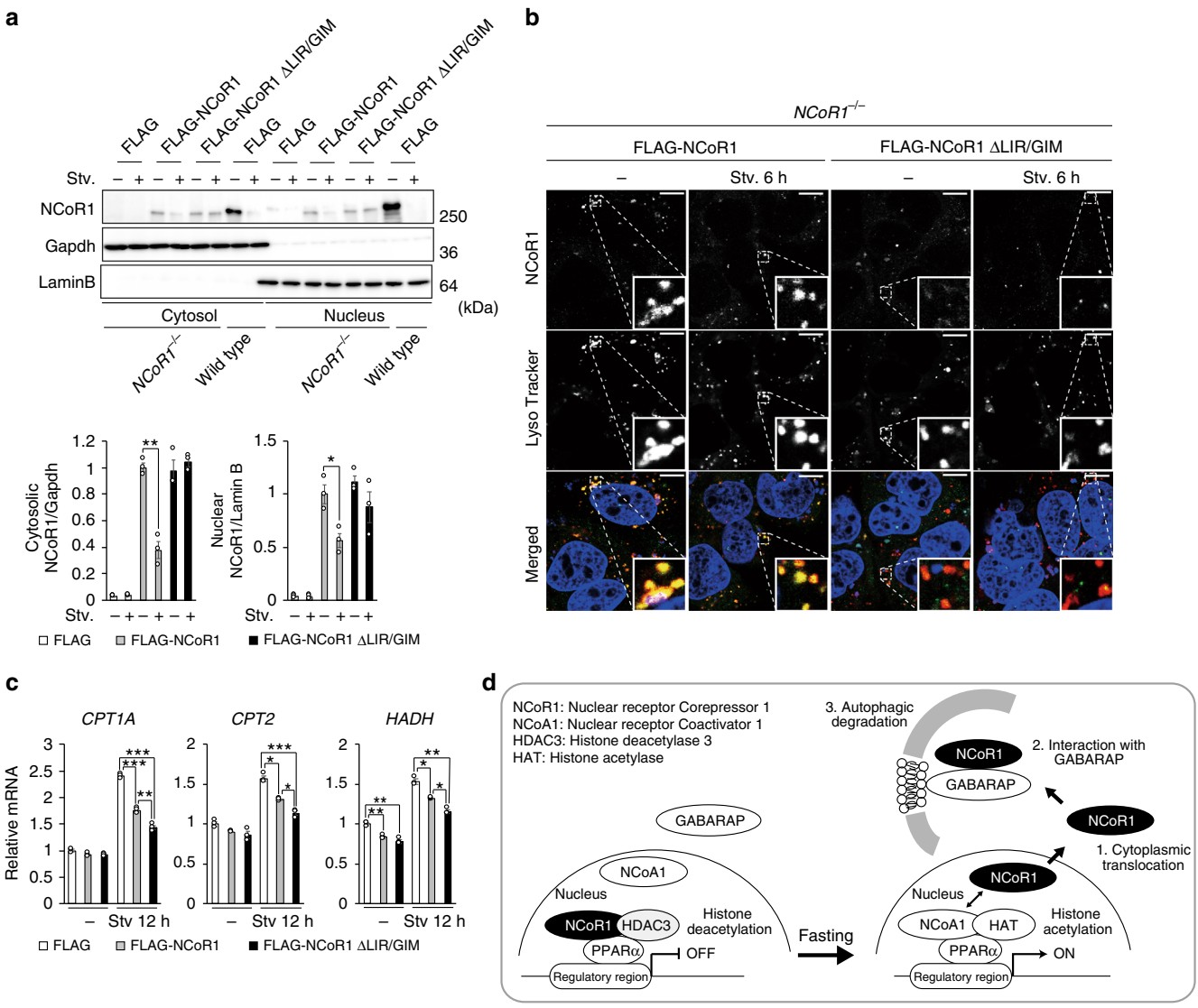

**Fig. 9** Autophagic degradation of NCoR1 dependent on the GABARAP-interaction. Immunoblot analysis. FLAG-NCoR1 or NCoR1 ΔLIR/GIM was expressed in *NCoR1*-knockout HepG2 cells. Forty-eight hours after transfection, the cells were cultured in nutrient-rich medium, or nutrient-deprived medium. Thereafter, cytoplasmic and nuclear fractions were prepared and subjected to immunoblotting with the indicated antibodies. Data are representative of three independent experiments. Bar graphs indicate the quantitative densitometric analyses of cytoplasmic and nuclear FLAG-NCoR1 and FLAG-NCoR1 ΔLIR/GIM relative to Gapdh and Lamin B, respectively. **b** Immunofluorescence analysis. FLAG-NCoR1 or NCoR1 ΔLIR/GIM was expressed in *NCoR1*-deficient HepG2 cells. Forty-eight hours after transfection, the cells were cultured in nutrient-rich medium, or nutrient-deprived medium for 6 h and then immunostained with anti-NCoR1 antibody. To label lysosomes, the cells were incubated with Lyso Tracker Deep Red (50 nM final) for 2 h before the immunostaining. **c** Real-time PCR analysis. Total RNAs were prepared from cells described in **a**. Values were normalized against the amount of mRNA in *NCoR1*-deficient HepG2 cells expressing FLAG. The experiments were performed three times. **d** Model of PPARα transactivation through selective autophagic degradation of NCoR1. Data are means ± s.e.m. *$P < 0.05$, **$P < 0.01$, and ***$P < 0.001$ as determined by Welch's *t*-test

$CaCl_2$, 2 mM $MgSO_4$, 1.2 mM $Na_2HPO_4$, 2.5 mM glucose, 0.5 mM carnitine, and 5 mM HEPES. Cells were incubated in a $CO_2$-free incubator at 37 °C for 45 min. An XF Instrument (Agilent Technologies, Santa Clara, CA, USA) gently mixed the assay media in each well for 10 min to allow the oxygen partial pressure to reach equilibrium. The oxygen consumption rate (OCR) was measured simultaneously three times to establish a baseline rate. For each measurement, with a total of 16 measurements, there was a 3-min mix followed by a 3-min wait time to restore normal oxygen tension and pH in the microenvironment surrounding the cells. Drug injection was performed throughout the assay. Etomoxir (40 μM final, Wako, E933505) was injected after measurements 4 and 8.

**Mice**. $p62^{-/-}$[36], $Atg5^{f/f}$[64], $Atg7^{f/f}$[4], $Atg5^{f/f}$;Mx1-*Cre*[36], $Atg7^{f/f}$;Alb-*Cre*[36], $Atg7^{f/f}$;$p62^{f/f}$;Alb-*Cre*[29], and $Atg7^{f/f}$;Nrf2^{f/f};Alb-*Cre*[48] mice in the C57BL/6 genetic background were used in this study. $NCoR1^{f/f}$ mice[37] were bred with $Atg7^{f/f}$;Alb-*Cre* and $Atg7^{f/f}$;$p62^{f/f}$;Alb-*Cre* mice to generate $Atg7^{f/f}$;NCoR $^{f/f}$;Alb-*Cre* and $Atg7^{f/f}$;$p62^{f/f}$;NCoR $^{f/f}$;Alb-*Cre* mice, respectively. To delete $Atg5$ in the liver, Cre expression was induced

in the liver by intraperitoneal injection of pIpC (Sigma Chemical, St. Louis, MO, USA). Mice were housed in specific pathogen-free facilities, and the Ethics Review Committee for Animal Experimentation of Niigata University, and of the University of Tokyo approved the experimental protocol. We have complied with all relevant ethical regulations. Blood glucose and β-hydroxybutyrate were measured using a glucose meter (Terumo, Tokyo, Japan) and blood ketone body meter (Abbott Laboratories, Chicago, IL, USA), respectively. The concentration of ketone bodies in plasma of genetically modified mice, except $Atg5^{f/f}$ and $Atg5^{f/f}$;Mx1-*Cre*, was determined using the EnzyChrom™ Ketone Body Assay Kit (EKBD-100, BioAssay Systems, Hayward, CA, USA). Free fatty acids in plasma were analyzed by SRL (Tokyo, Japan).

**Immunoblot analysis**. Livers were homogenized in 0.25 M sucrose, 10 mM 2-[4-(2-hydroxyethyl)-1-piperazinyl]ethanesulfonic acid (HEPES) (pH 7.4), and 1 mM dithiothreitol (DTT). Nuclear and cytoplasmic fractions from livers and cultured cells were prepared using the NE-PER Nuclear and Cytoplasmic Extraction

Reagents (Thermo Fisher Scientific). Samples were separated using the NuPAGE system (Life Technologies, Grand Island, NY, USA) on 4–12% Bis–Tris gels in MOPS-SDS buffer, and then transferred to a polyvinylidene difluoride (PVDF) membrane. Antibodies against PPARα (ab8934, Abcam, Cambridge, UK; 1:500), NCoR1 (#5948S, Cell Signaling Technology, Danvers, MA, USA; 1:500), Atg7 (013-22831, Wako Pure Chemical Industries, Osaka, Japan; 1:1000), p62 (GP62-C, Progen Biotechnik GmbH, Heidelberg, Germany; 1:1000), LC3B (#2775, Cell Signaling Technology; 1:500), ULK1 (sc-33182, Santa Cruz Biotechnology, Dallas, TX, USA; 1:1000), S6K (#2708, Cell Signaling Technology; 1:500), T421/S424-phosphorylated S6K (#9204S, Cell Signaling Technology; 1:500), Gapdh (MAB374, Merck Millipore Headquarters, Billerica, MA, USA; 1:1000), Actin (MAB1501R, Merck Millipore Headquarters; 1:1000), Lamin B (M-20, Santa Cruz Biotechnology; 1:200), and FLAG (185-3L, Medical and Biological Laboratories, Nagoya, Japan; 1:1000) were purchased from the indicated suppliers. Blots were then incubated with horseradish peroxidase-conjugated secondary antibody (Goat Anti-Mouse IgG (H + L), 115-035-166, Jackson ImmunoResearch; Goat Anti-Rabbit IgG (H + L) 111-035-144; Goat Anti-Guinea Pig IgG (H + L) 106-035-003; Donkey Anti-Goat IgG (H + L) 705-035-003; all 1:10,000) and visualized by chemiluminescence. Full size images are presented in Supplementary Figure 11–15.

**Pull-down analysis.** One-Strep-FLAG (OSF)-tagged LC3B or individual GABARAP family proteins were expressed alone or together with FLAG-tagged NCoR1 or NCoR1 ΔLIR/GIM in HEK293T cells. Forty-eight hours after transfection, the cells were lysed with TNE buffer (20 mM Tris-Cl, pH 7.5, 0.5% Nonidet P-40, 150 mM NaCl, 1 mM ethylenediaminetetraacetic acid [EDTA]), and 1 mM DTT containing protease inhibitor cocktail (Roche Applied Science). The cell lysates were then incubated with Strep-Tactin Sepharose (IBA, GmbH, Gottingen, Germany) at 4 °C for 2 h. Alternatively, OSF-GABARAP immobilized on Strep-Tactin Sepharose was mixed with lysates prepared from cells expressing each NCoR1-deletion mutant, and then incubated at 4 °C for 2 h. The pulled-down protein complexes were collected by centrifugation and extensively washed with TNE buffer containing 1.0 M NaCl. The resultant precipitants were analyzed by immunoblotting.

**qRT-PCR (quantitative real-time PCR).** Using the Transcriptor First-Strand cDNA Synthesis Kit (Roche Applied Science, Indianapolis, IN, USA), cDNA was synthesized from 1 μg of total RNA. Quantitative PCR was performed using the LightCycler® 480 Probes Master mix (Roche Applied Science) on a LightCycler® 480 (Roche Applied Science). Signals from human and mouse samples were normalized against *GAPDH* (glyceraldehyde-3-phosphate dehydrogenase) and *Gusb* (β-glucuronidase), respectively. The sequences of the primers used for analysis of mouse livers or human cell lines are provided in Supplementary Table 1.

**Histological experiments.** Excised liver tissues were fixed in 4% paraformaldehyde and embedded in OCT-compound. For immunofluorescence microscopy, cryosections were processed for antigen retrieval for 20 min at 98 °C using a microwave processor (MI-77, AZUMAYA, Japan) in 1% Immunosaver (Nissin EM, Japan). For immunofluorescence staining, sections were incubated for 2 to 3 days at 4 °C with rabbit polyclonal antibody against NCoR1 (#5948 S, Cell Signaling Technology). Alexa Fluor 488-conjugated donkey anti-rabbit IgG (Invitrogen, San Diego, CA, USA) was used as secondary antibody. The sections were counterstained with Hoechst33342, and observed with a laser-scanning confocal microscope (FV1000, Olympus) equipped with a ×40 objective lens (UPlanSApo, oil, NA 1.3, Olympus). After image acquisition, contrast and brightness were adjusted using Photoshop CS4 (Adobe Systems, San Jose, CA, USA). To examine lipid droplets, the sections were also stained with Oil Red O, and then observed with a microscope (BX51, Olympus).

**Immunofluorescence microscopy for cultured cells.** Cells grown on coverslips were fixed in 4% paraformaldehyde in PBS for 15 min, permeabilized with 0.1% Triton X-100 in PBS for 5 min, blocked with 0.1% (w/v) gelatin (Sigma-Aldrich) in PBS for 30 min, and then incubated overnight with primary antibodies. For Lyso Tracker staining, cells were incubated with Lyso Tracker Deep Red (L12492, Invitrogen) for 2 h. Antibodies against NCoR1 (#5948 S, Cell Signaling Technology; dilution ratio is 1:200), GABARAP (M135-3, Medical and Biological Laboratories, Nagoya, Japan; dilution ratio is 1:200), and Lamp1 (H4A3, Santa Cruz Biotechnology; dilution ratio is 1:200) were used as primary antibodies. After washing, cells were incubated with Goat anti-Rabbit IgG (H + L) Cross-Adsorbed Secondary Antibody, Alexa Fluor 488 (A11008, Thermo Fisher Scientific, Waltham, MA, USA; dilution ratio is 1:1000), and Goat anti-Mouse IgG (H + L) Highly Cross-Adsorbed Secondary Antibody, Alexa Fluor 647 (A21236, Thermo Fisher Scientific, Waltham, MA, USA; dilution ratio is 1:1000) for 60 min. Cells were imaged using a confocal laser-scanning microscope (Olympus, FV1000) with a UPlanSApo ×100 NA 1.40 oil objective lens. Z-projection stack images were acquired with z steps of 0.5 μm. Image contrast and brightness were adjusted using Photoshop CS4 (Adobe System).

**ChIP coupled with quantitative PCR.** Chromatin immunoprecipitation (ChIP) assay was performed with an anti-H3K27ac antibody (MABI0309, MAB Institute, Inc., Sapporo, Japan) as previously described[65]. Wild-type and *ATG7*-knockout HepG2 cells were cross-linked with 1% formaldehyde for 10 min, and the lysed chromatin was fragmented by sonication. Solubilized chromatin was incubated overnight with Dynabeads anti-mouse IgG (Invitrogen) prebound with control IgG or anti-H3K27ac antibody. The precipitated DNA was PCR-amplified using primers listed in Supplementary Table 2. Enhancer regions with H3K27ac deposition for the PCR amplification were selected from the *CPT1A* and *CPT2* loci based on HepG2 cell data in ENCODE database.

**Lipidome analysis.** Lipidome analysis was performed as described previously with slight modification[66]. Tissues (15–45 mg) were crushed to powder with an SK mill (SK-100; Tokken, Chiba, Japan) without thawing, and lipids were extracted by the method of Bligh and Dyer[67] with internal standards. The organic (lower) phase was transferred to a clean vial and dried under a stream of nitrogen. The lipids were then resolubilized in methanol containing 0.1% (v/v) formic acid, and a portion of the extracted lipid was injected onto the liquid chromatography/tandem mass spectrometry (LC–MS/MS) system. LC separation was performed on an ACQUITY UPLC™ BEH $C_{18}$ column (1.7 μm, 2.1 mm × 100 mm; Waters, Milford, MA, USA) coupled to an ACQUITY UPLC™ BEH $C_{18}$ VanGuard™ Pre-column (1.7 μm, 2.1 mm × 5 mm; Waters). Mobile phase A was 60:40 (v/v) acetonitrile:$H_2O$ containing 10 mM ammonium formate and 0.1% (v/v) formic acid, and mobile phase B was 90:10 (v/v) isopropanol:acetonitrile containing 10 mM ammonium formate and 0.1% (v/v) formic acid. LC gradient consisted of 20% B for 2 min, a linear gradient to 60% B over 4 min, a linear gradient to 100% B over 16 min, and equilibration with 20% B for 5 min (total run time of 27 min). The flow rate was 0.3 mL/min, and column temperature was 55 °C. Multiple reaction monitoring (MRM) was performed using a Xevo™ TQ-S micro triple-quadrupole mass spectrometry system (Waters) equipped with an electrospray ionization (ESI) source. ESI capillary voltage was set at 1.0 kV, and the sampling cone was set at 30 V. The source temperature was set at 150 °C, desolvation temperature was set at 500 °C, and desolvation gas flow was 1000 L/h. The cone gas flow was set at 50 L/h.

**Tracer experiments with [$^{13}$C] palmitate.** Mice were fasted for 24 h. For experiments tracing the metabolic fate of palmitate, mice received $^{13}C_{16}$-palmitate (2.4 μg/body weight g) by intravenous administration 60 min before sacrifice under anesthesia. A portion of the liver lobule was sampled for snap-freezing using liquid nitrogen. Collected liver tissues were stored −80 °C until use. Metabolite extraction from tissues for metabolome analyses was performed as described previously[68]. Frozen livers together with internal standard (IS) compounds (2-morpholi-noethanesulfonic acid (MES) and 1,3,5-benzenetricarboxylic acid (trimesate)) was homogenized in ice-cold methanol (500 μL) using a manual homogenizer (Finger Masher (AM79330), Sarstedt), followed by the addition of an equal volume of chloroform and 0.4 volumes of ultrapure water (LC/MS grade). The suspension was then centrifuged at 15,000 × g for 15 min at 4 °C. After centrifugation, the aqueous phase was ultrafiltered using an Ultrafree MC-PLHCC ultrafiltration tube (Human Metabolome Technologies). The filtrate was concentrated on a vacuum concentrator (SpeedVac, Thermo). The concentrated filtrate was dissolved in 50 μL of ultrapure water and subjected to IC-MS analyses. For metabolome analysis focused on ketone body production from administered $^{13}C_{16}$-palmitate, anionic metabolites were measured using an Orbitrap-type MS (Q-Exactive focus, Thermo Fisher Scientific) connected to an high performance ion-chromatography system (ICS-5000+, Thermo Fisher Scientific) that enabled us to perform highly selective and sensitive metabolite quantification owing to the IC-separation and Fourier Transfer MS principle[69]. The IC was equipped with an anion electrolytic suppressor (Thermo Scientific Dionex AERS 500), which converted the potassium hydroxide gradient into pure water before the sample entered the mass spectrometer. Separation was performed using a Thermo Scientific Dionex IonPac AS11-HC, 4-μm particle size column. The IC flow rate was 0.25 mL/min supplemented post-column with 0.18 mL/min makeup flow of MeOH. The potassium hydroxide gradient conditions for IC separation are as follows: from 1 mM to 100 mM (0–40 min), 100 mM (40–50 min), and 1 mM (50.1–60 min), at a column temperature of 30 °C. The Q-Exactive focus mass spectrometer was operated under ESI-negative mode for all detections. Full mass scan (m/z 70–900) was used at a resolution of 70,000. The automatic gain control target was set at 3 × 10⁶ ions, and maximum ion injection time was 100 ms. Source ionization parameters were optimized with the spray voltage at 3 kV, and other parameters were as follows: transfer temperature at 320 °C, S-Lens level at 50, heater temperature at 300 °C, Sheath gas at 36, and Aux gas at 10.

**Statistical analysis.** Values, including those displayed in the graphs, are means ± s.e.m. Statistical analysis was performed using the unpaired *t*-test (Welch test) (two-sided). A *P*-value <0.05 was considered to indicate statistical significance.

## Data availability

The authors declare that the data supporting the findings of this study are available within the article and its supplementary information. A reporting summary for this

article is available as a supplementary information file. Source data for Fig. 1b and Supplementary Figs. 1 and 2 is provided as a Source Data file and was deposited in figshare (https://figshare.com/articles/Source_data_docx/7575824).

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

## Acknowledgements

We thank T. Kouno (Niigata University) and Y. Yoshimura (The University of Tokyo) for excellent technical assistance, K. Kanno and H. Annoh (Fukushima Medical University School of Medicine) for their help with histological studies, and M. Sugimoto (Keio University) for help with data processing for the comprehensive lipidome analysis. We also thank M.S. Lee (Yonsei University) for critical reading and comments on this manuscript. T.S. is supported by a Grant-in-Aid for JSPS Research Fellows (17J05623). A.K. is supported by JSPS PRESTO. M.K. and N.M. are supported by Grants-in-Aid for Scientific Research on Innovative Areas (JP25111006 to M.K, and JP25111005 to N.M.), the Japan Society for the Promotion of Science (an A3 foresight program, to M.K., 15H06600 to H.-C.L, 15H04708, 15H05879, and 15H05904 to T.Y.), and the Takeda Science Foundation (to M.K.). J.A. is supported by the EPFL and the Swiss National Science Foundation (31003A-140780).

## Author contributions

M.K. designed and directed the study. M.K., T. Saito, A.K. and N.M. wrote the manuscript. T. Saito, A.K., Y.S., Y.I. and M.O. carried out the biochemical and cell biological experiments. H.K. and H.M. carried out the ChIP coupled with qPCR. T. Saito and A.K. performed most of the experiments that characterized the knockout mice. S.W. performed histological and microscopic analyses. H.-C.L., K.I., M.S., T. Soga., and T.Y. performed lipidome analyses. S.O. performed bioinformatic analyses. Y.K. and I.S. developed adenovirus for expression of NCoR1. J.A. provided *NCoR1* conditional knockout mice. All authors discussed the results and commented on the manuscript.

## Additional information

**Competing interests:** The authors declare no competing interests.

**Journal Peer Review Information:** *Nature Communications* thanks the anonymous reviewers for their contribution to the peer review of this work.

