## [Peer Review File · Nature Communications]

Reviewers' comments:

Reviewer #1 (Remarks to the Author):

Saito et al. addressed the fascinating field of autophagy and its role on the regulation of lipid oxidation by controlling NCoR1 turnover. In this paper they show that the ablation of *atg7* and *atg5*, which are essential for autophagy, impaired beta-oxidation in the liver of mice. This was associated with decreased translocation of acylcarnitines into the mitochondria, and the consequent increase of acylcarnitines in both the fed and fasted condition. This in turn was associated with decreased generation of BHB during fasting, and was consistent with decreased expression of carnitine transferases, *Lcad* and *Cact*, which was explained by decreased expression of *Pparalpha*. Also, *Nrf2* target genes were upregulated in response to increased p62 in *atg7KO* as shown in previous reports. The authors then looked into the expression of NCoR1 which represses *Pparalpha* and its action and found increased levels of NCoR1 protein but not RNA in the *atg7KO* mice, in both fed and fasted conditions which correlated with decreased levels of *Pparalpha* protein. Also, they showed that the ablation of NCoR1 prevented the downregulation *pparalpha* targets, in *atg7KO* mice. The authors then developed a triple KO mice where they ablated the expression of *atg7*, p62, and NCoR1 together, and showed rescued expression of *CpT1*, *CpT2*, *Lcad* and *Cact*, and prevented upregulation of *Nrf2* targets. This in turn allowed the authors to hypothesize that the decrease in BHB generation in *atg7KO* mice is due to NCoR1 accumulation and repression of *Pparalpha* targets. Finally they discovered and characterized a novel interaction between NCoR1 and GABARAP, which suggested that autophagy decreases NCoR1 during fasting to facilitate fatty acid oxidation.

Overall this paper is very interesting and some new information for the field.

Comments:

- a) Figure 1 shows increased levels of acetylcarnitines which can also be related to higher levels of fatty acid oxidation, here the authors indicate decreased fatty acid oxidation in *Atg7* and *5ko* mice, this discrepancy must be clarified in the paper
- b) The authors showed decreased generation of BHB in fasting *Atg7KO* and *Atg5KO* mice, and concluded that this is due to NCoR1 accumulation, however no data of BHB synthesis in fasted *Atg7KO/NCoR1* or *Atg7/p62/NCoR1* mice are shown
- c) figure 2, 3, and 4 show no NCoR1 band on western blots in control mice, however mRNA levels are the same across models in different conditions. How does the lack of NCoR1 band in fed control mice would support the hypothesis that diminished NCoR1 protein is needed to fully activate lipid oxidation by *pparalpha*? Also, the accumulation of NCoR1 in the nucleus of *Atg7KO* mice is higher than in the cytosol, suggesting an increase in the translocation from cytosol to nucleus in autophagy impaired mice, which is consistent with the autophagy-independent regulation of NCoR1 shown by Sinha et al (AUTOPHAGY 2017, VOL. 13, NO. 1, 169–186).
- d) Specificity across the manuscript would be helped by including assessment of the protein levels of SMRT in some of the conditions tested.
- e) figure 2c, 4c immunofluorescence images have poor quality and are very difficult to interpret,
- f) supplemental figure 4a shows decreased expression of fatty acid oxidation genes which contradicts the conclusion that NCoR1 is needed to repress these genes in *atg7ko* mice, did the author looked at the expression of other corepressors e.g. SMRT?
- g) In figure 5 and 6 the authors indicate that NCoR1 interaction with GABARAP is needed for autophagic mediated turnover of NCoR1. Does the ablation of FxxI motif in NCoR1 recapitulates *atg7KO* phenotype in context of fatty acid oxidation?
- h) A major conclusion of this work is that the accumulation of NCoR1 in *atg7KO* mice is responsible for decreased gene expression of beta-oxidation regulating genes, This must be supported by experimental data showing the increase of NCoR1 recruitment to the regulatory regions of these genes or the presence of less acetylation marks.
- i) Experiments in cell culture alternate between *hepG2* and *hek293t* cell lines, this inconsistency

should be clarified in the manuscript

j) in *AUTOPHAGY* 2017, VOL. 13, NO. 1, 169–186, Sinha et al. showed that activation of ULK1 leads to decreased levels of NCoR1 in the nucleus by decreasing of RPS6KB1 activity, which in turn upregulates SCD1 to prevent lipotoxicity in an autophagy independent mechanism. How does the nature of the lipids measured in the present paper correlate with this conclusion? does the ratio of saturated/unsaturated fatty acids ultimately impact the rate of lipid oxidation and generation of ketone bodies?. Also, the use of commercial ligands for Pparalpha in Atg7Ko mice could potentially support the notion that NCoR1 is repressing fatty acid oxidation

Reviewer #2 (Remarks to the Author):

This manuscript reports the study showing that autophagy could act through NCoR1 to regulate hepatic lipid metabolism. Using several genetic knockout mouse models as well as cultured cells, the presented results suggest that autophagy is directly involved in mediating the degradation of NCoR1, and defective autophagy leads to increased NCoR1 and thus inhibition of fatty acid oxidation. The authors concluded that "autophagy contributes to full activation of PPAR α upon fasting by promoting degradation of its repressor, NCoR1 and is involved in β -oxidation and subsequent production of ketone bodies".

Overall this is a well-conceived study with interesting mechanistic data, and the experiments were well conducted. The results are clearly presented, which will further our molecular understanding of the metabolic role of autophagy-related control of metabolic pathways in hepatic lipid homeostasis.

The major issues that need to be clarified are as follows. The presented data showed that liver ATG7 KO mice had elevated levels of acylcarnitines with lower expression of β -oxidation genes, suggesting a reduced lipid oxidation. How this goes along with the impact of autophagy deficiency upon hepatosteatosis during fasting is unclear. Previously published results reported decreased versus increased fasting-induced hepatic lipid accumulation in liver ATG7 LKO mice (Kim KH, et al. *Nat Med.* 2013 Jan; 19(1): 83-92, Figure 6 and Figure S10; Singh R, et al. *Nature* 458, 1131-1135 (2009)). This raises the question as to how important the current finding is regarding autophagy-associated NCoR1 regulation in the liver in vivo. In addition, more direct evidence appears to be desirable for supporting autophagy-mediated degradation of NCoR1 in the fasted state. Whether autophagy deficiency leads to mitochondrial dysfunction and subsequently impairs lipid utilization remains an open question.

Specific points:

1. Given the possibility that the observed reduction in hepatic lipid oxidation in liver ATG7 KO mice might arise from defective mitochondrial autophagy and mitochondrial dysfunction, it would be helpful to examine the functional changes of mitochondria (e.g. by electron microscopy, etc) in relation to NCoR1 regulation.
2. In addition to the immunofluorescent results in Fig 5e, more evidence is needed to support autophagy-mediated degradation of NCoR1, e.g. the co-localization of NCoR1/GABARAP puncta with a lysosome marker such as LAMP2 in response to fasting, or additional markers and changes in their co-localization with NCoR1 in LC3-GFP-expressing cells, or even with endogenous LC3 co-localization if technically feasible. Immunogold electron microscopy with anti-NCoR1 antibody may be another useful strategy.
3. Are there differences in lipid contents or fatty acid oxidation levels between Atg7-p62 double KO and Atg7-p62-NCoR1 triple KO livers under fed versus fasted states?
4. In Fig 2b, it appears that nuclear NCoR1 levels dramatically decreased upon fasting in liver ATG7-deficient mice, but this did not happen in liver ATG7-p62 double KO mice in Fig 4b. Does this suggest p62 has a role in regulating NCoR1 during fasting in the absence of autophagy? How come

p62 protein was not detectable in the control mice in Fig 4b? Fasting induction of PPARalpha was not consistently observed in these different models. All immunoblots from multiple replicates of the experiments should be quantified to show statistical significance.

5. For immunostaining of NCoR1 in Fig 2c and 4c, the quality of fluorescent images can be improved, perhaps using nuclear markers with color images.

6. For Supplementary Table S1, the data might be better presented if changes were shown as heat maps.

Reviewer #3 (Remarks to the Author):

The authors identified a novel autophagy-dependent mechanism that control lipid homeostasis. This study started after a lipidomic analyses on liver of conditional Atg7 knockout that showed an accumulation of Acylcarnitine species. This finding together with a decrease of ketone bodies in the blood suggest an impairment in lipid oxidation. This was the rationale to look at PPARa, a critical nuclear receptor that controls lipid metabolism and found that was decreased because the inhibitor NCoR was dramatically upregulated. Gain and loss of function experiments in vivo and in vitro including also rescue experiments confirmed that NCoR was indeed a substrate of autophagy and was the culprit of PPARa inhibition and of the lipid metabolism reduction. Finally, they also found a LIR/GIR that allows the NCoR binding to Gabarap protein on the autophagosome. The paper is well written the experiments are well designed and data are consistent with author hypothesis and conclusion. This is an excellent paper with many in vivo experiments and transgenic/knockout mice. The story is novel and would clarify several missing points and contradictory data on autophagy and regulation of lipids. I have only two minor points.

1) In nuclear and cytosolic fraction western blots, actin should be checked in nuclear and Lamin in cytosolic fraction to show that there is no contamination.

2) Does the NCoR/Gabarab colocalization increase after lysosomal inhibition (BafA)? And which is the explanation of the increase Gabarap/NCoR colocalization when ATG7 is absent? The expectation is a reduction of the colocalization when the conjugation system is blocked.

3) The authors showed in Fig2a a decrease in PPARa transcript, is this repression consequent to NCoR upregulation or is independent of NCoR

Reviewers' Comments: (italicized)

Reviewer #1:

General comments

Saito et al. addressed the fascinating field of autophagy and its role on the regulation of lipid oxidation by controlling NCoR1 turnover. In this paper they show that the ablation of atg7 and atg5, which are essential for autophagy, impaired beta-oxidation in the liver of mice. This was associated with decreased translocation of acylcarnitines into the mitochondria, and the consequent increase of acylcarnitines in both the fed and fasted condition. This in turn was associated with decreased generation of BHB during fasting, and was consistent with decreased expression of carnitine transferases, Lcad and Cact, which was explained by decreased expression of Pparalpha. Also, Nrf2 target genes were upregulated in response to increased p62 in atg7KO as shown in previous reports. The authors then looked into the expression of NCoR1 which represses Pparalpha and its action and found increased levels of NCoR1 protein but not RNA in the atg7KO mice, in both fed and fasted conditions which correlated with decreased levels of Pparalpha protein. Also, they showed that the ablation of NCoR1 prevented the downregulation pparalpha targets, in atg7KO mice. The authors then developed a triple KO mice where they ablated the expression of atg7, p62, and NCoR1 together, and showed rescued expression of CpT1, CpT2, Lcad and Cact, and prevented upregulation of Nrf2 targets. This in turn allowed the authors to hypothesize that the decrease in BHB generation in atg7KO mice is due to NCoR1 accumulation and repression of Pparalpha targets. Finally they discovered and characterized a novel interaction between NCoR1 and GABARAP, which suggested that autophagy decreases NCoR1 during fasting to facilitate fatty acid oxidation.

Overall this paper is very interesting and some new information for the field.

Reply

We greatly appreciate the overall evaluation.

Comment-1

Figure 1 shows increased levels of acetylcarnitines which can also be related to higher levels of fatty acid oxidation, here the authors indicate decreased fatty acid oxidation in Atg7 and 5ko mice, this discrepancy must be clarified in the paper.

Reply-1

The impairment of mitochondrial β -oxidation increases the ratio of acylcarnitine/carnitine (Reuter SE and Evans AM, *Clin Pharmacokinet.*, 51, 553-572, 2012). In Figure 1, we showed that while the level of acylcarnitines in livers of *Atg7^{fl/fl}; Alb-Cre* and *Atg5^{fl/fl}; Mx1-Cre* mice increases under both fed and fasting conditions, the level of carnitine decreases under fasting conditions, implying defective β -oxidation. We cited the paper by Reuter SE and Evans AM in the revised manuscript (line 142-143 in the revised manuscript).

Comment-2

The authors showed decreased generation of BHB in fasting Atg7KO and Atg5KO mice, and concluded that this is due to NCoR1 accumulation, however no data of BHB synthesis in fasted Atg7KO/NCoR1 or Atg7/p62/NCoR1 mice are shown.

Reply-2

In accordance to this suggestion, we measured the blood level of BHB in liver-specific *Atg7 p62 NCoR1*-triple knockout mice and verified the recovery of decreased level of blood BHB observed in liver-specific *Atg7*- and *Atg7 p62*-knockout mice (Figure 4d in the revised manuscript).

Comment-3

figure 2, 3, and 4 show no NCoR1 band on western blots in control mice, however mRNA levels are the same across models in different conditions. How does the lack of NCoR1 band in fed control mice would support the hypothesis that diminished NCoR1 protein is needed to fully activate lipid oxidation by PPAR α ?

Reply-3

As the reviewer pointed out, we did not detect endogenous NCoR1 in liver homogenates of control *Atg7^{fl/fl}* mice, probably due to technical difficulties. Instead, we conducted the experiments with mouse primary cultured hepatocytes isolated from *Atg7^{fl/fl}* mice. As shown in Figure 2c in the revised manuscript, NCoR1 protein was clearly detected in both nuclear and cytoplasmic fractions of the primary cultured hepatocytes, and NCoR1 in both fractions decreased upon ketogenic conditions, which was not observed in the case of *Atg7*-deletion. Gene expression of PPAR α targets was inversely correlated to NCoR1 level (Figure 2d in the revised manuscript). We described the above-mentioned modifications in the result section of the revised manuscript (line 175-181 in the revised manuscript).

Comment-4

Also, the accumulation of NCoR1 in the nucleus of *Atg7KO* mice is higher than in the cytosol, suggesting an increase in the translocation from cytosol to nucleus in autophagy impaired mice, which is consistent with the autophagy-independent regulation of NCoR1 shown by Sinha *et al* (AUTOPHAGY 2017, VOL. 13, NO. 1, 169–186).

Reply-4

Sinha *et al* showed that phosphorylation of RPS6KB1 induced by knockdown of *ULK1* promotes nuclear translocation of NCoR1. We examined the phosphorylation level of RPS6KB1 in *Atg7*-deficient primary cultured hepatocytes and found that the level was comparable to that in control hepatocytes. We cited the paper (Autophagy, 13, 169-186, 2017) and explained that the nuclear accumulation of NCoR1 in *Atg7*-deficient hepatocytes depends on a different mechanism from the RPS6KB1-mediated translocation of NCoR1 (line 375-380 in the revised manuscript).

Immunoblot analysis. Primary hepatocytes were isolated from *Atg7^{fl/fl}* mice and infected with adenovirus for GFP or Cre recombinase. Seventy-two hours after infection, the cells were cultured under nutrient-rich and ketogenic conditions for 24 hr, and then cytoplasmic fraction was prepared from the cells and subjected to immunoblotting with the indicated antibodies. Data are representative of three separate experiments. Bar graphs indicate the quantitative densitometric analyses of cytoplasmic p-RPS6KB1 relative to Gapdh. Statistical analyses were performed using Welch's *t*-test. Data are means \pm s.e. ***P* < 0.01, and ****P* < 0.001 as determined by Welch's *t*-test.

Comment-5

Specificity across the manuscript would be helped by including assessment of the protein levels of SMRT in some of the conditions tested.

Reply-5

In accordance with the reviewer's comment, we conducted the immunoblot analysis with anti-SMRT antibody, but we could not detect clear signal. Instead, according to the comment 4 of the reviewer 2, we quantified all immunoblot data shown in revised manuscript. The specific accumulation of NCoR1 in *Atg7*-deficient mouse livers was ensured by the quantification.

Comment-6

figure 2c, 4c immunofluorescence images have poor quality and are very difficult to interpret.

Reply-6

According to this suggestion together with comment 5 of the Reviewer 2, we retried the immunofluorescence analysis. As shown in Figure 2e and 4c in the revised manuscript, the quality was significantly improved.

Comment-7

*supplemental figure 4a shows decreased expression of fatty acid oxidation genes which contradicts the conclusion that NCoR1 is needed to repress these genes in *atg7ko* mice, did the author looked at the expression of other corepressors e.g. SMRT?*

Reply-7

As the reviewer pointed out, gene expression of some PPAR α targets in *Atg7 NCoR1*-double knockout mouse livers still remained to be recovered (Supplementary Figure S6a). But, in *Atg7 p62 NCoR1*-triple knockout mouse livers, the gene expression was completely recovered (Figure 4a). Since liver phenotypes in the double knockout mice were quite severer than those in the triple knockout mice (Supplementary Figure S5a and b), the liver damages seem to affect the gene expression. In support of this hypothesis, suppression of gene expression of PPAR α targets in HepG2 cells lacking *ATG7* was completely recovered by knockdown of only NCoR1 (Figure 3f and g). We have stated the above-mentioned matters in the original manuscript.

Comment-8

*In figure 5 and 6 the authors indicate that NCoR1 interaction with GABARAP is needed for autophagic mediated turnover of NCoR1. Does the ablation of FxxI motif in NCoR1 recapitulates *atg7KO* phenotype in context of fatty acid oxidation?*

Reply-8

Thank you for this valuable suggestion. We investigated whether introduction of GIM-deleted NCoR1 into *NCoR1*-knockout HepG2 cells recapitulates suppression of gene expression of PPAR α targets that is observed in *Atg7*-deficient conditions or not. As shown in Figure 7a-c in the revised manuscript, we confirmed that GIM-deleted NCoR1 shows resistance to starvation-induced degradation and has a suppressive effect on gene expression of PPAR α targets (Figure 7a-c in the revised manuscript). We described the results in the result section of the revised manuscript (line 309-320 in the revised manuscript).

Comment-9

A major conclusion of this work is that the accumulation of NCoR1 in atg7KO mice is responsible for decreased gene expression of beta-oxidation regulating genes, This must be supported by experimental data showing the increase of NCoR1 recruitment to the regulatory regions of these genes or the presence of less acetylation marks.

Reply-9

Thank you for this valuable comment again. We conducted Chromatin immunoprecipitation (ChIP) with anti-acetylated lysine 27 of histone H3 (H3K27ac) coupled with quantitative PCR. As shown in Figure 3c, H3K27ac deposition on promoter regions of PPAR α targets was significantly and specifically decreased by loss of *ATG7*, which is consistent with the function of NCoR1 as a transcription corepressor (Figure 3c in the revised manuscript). We described the result in the result section of the revised manuscript (line 195-203 in the revised manuscript).

Comment-10

Experiments in cell culture alternate between hepG2 and hek293t cell lines, this inconsistency should be clarified in the manuscript.

Reply-10

We used HEK293T cells for the simple interaction analysis. We stated this in the result section (line 290-291 in the revised manuscript).

Comment-11

in AUTOPHAGY 2017, VOL. 13, NO. 1, 169–186, Sinha et al. showed that activation of ULK1 leads to decreased levels of NCoR1 in the nucleus by decreasing of RPS6KB1 activity, which in turn upregulates SCD1 to prevent lipotoxicity in an autophagy independent mechanism. How does the nature of the lipids measured in the present paper correlate with this conclusion? does the ratio of saturated/unsaturated fatty acids ultimately impact the rate of lipid oxidation and generation of ketone bodies?.

Reply-11

In accordance to this comment, we checked the levels of saturated fatty acids and of unsaturated fatty acids and found that the result is consistent with the data presented in Journal Autophagy (vol. 13, No. 1, 169-186 2017), raising a possibility that the model proposed by Sinha *et al* may affect the decreased ketogenesis in *Atg7*-deficient mouse livers. We cite this paper and discuss the above-mentioned considerations in the discussion section (line 380-387 in the revised manuscript).

Comment-12

Also, the use of commercial ligands for Pparalpha in Atg7Ko mice could potentially support the notion that NCoR1 is repressing fatty acid oxidation.

Reply-12

We administrated an agonist of PPAR α , WY-14643 into *Atg5^{flox/flox}* and *Atg5^{flox/flox}; Mx1-Cre* mice. Though preliminary data, while gene expression of PPAR α targets in control mouse livers was induced by the treatment of WY-14643, that was not the case in mutant livers, suggesting that the accumulation of NCoR1 is a primary cause for suppressing β -oxidation.

Expression of genes encoding enzymes related to lipid oxidation. *Atg5^{f/f}* and *Atg5^{f/f};Mx1-Cre* mice were intraperitoneally injected with plpC to delete *Atg5* in the liver at the age of 10 weeks. At the age of 12 weeks, they were administrated orally by corn oil (-) or WY-14643 (20 mg/kg body weight) for 4 days. Total RNAs were prepared from livers of the mice (n = 3). Values were normalized against the amount of mRNA in the liver of plpC-injected and corn oil-administrated *Atg5^{f/f}* mice. Experiments were performed three times. Data are means \pm s.e. * $P < 0.05$ and ** $P < 0.01$ as determined by Welch's *t*-test.

Reviewer #2 (Remarks to the Author):

General comments

This manuscript reports the study showing that autophagy could act through NCoR1 to regulate hepatic lipid metabolism. Using several genetic knockout mouse models as well as cultured cells, the presented results suggest that autophagy is directly involved in mediating the degradation of NCoR1, and defective autophagy leads to increased NCoR1 and thus inhibition of fatty acid oxidation. The authors concluded that "autophagy contributes to full activation of PPAR α upon fasting by promoting degradation of its repressor, NCoR1 and is involved in β -oxidation and subsequent production of ketone bodies".

Overall this is a well-conceived study with interesting mechanistic data, and the experiments were well conducted. The results are clearly presented, which will further our molecular understanding of the metabolic role of autophagy-related control of metabolic pathways in hepatic lipid homeostasis.

The major issues that need to be clarified are as follows. The presented data showed that liver ATG7 KO mice had elevated levels of acylcarnitines with lower expression of β -oxidation genes, suggesting a reduced lipid oxidation. How this goes along with the impact of autophagy deficiency upon hepatosteatosis during fasting is unclear. Previously published results reported decreased versus increased fasting-induced hepatic lipid accumulation in liver ATG7 LKO mice (Kim KH, et al. Nat Med. 2013 Jan; 19(1): 83-92, Figure 6 and Figure S10; Singh R, et al. Nature 458, 1131-1135 (2009)). This raises the question as to how important the current finding is regarding autophagy-associated NCoR1 regulation in the liver in vivo. In addition, more direct evidence appears to be desirable for supporting autophagy-mediated degradation of NCoR1 in the fasted state. Whether autophagy deficiency leads to mitochondrial dysfunction and subsequently impairs lipid utilization remains an open question.

Reply of general comments

We greatly appreciate the overall positive evaluation and thank the reviewer for their helpful comments. The reviewer had an interest in whether NCoR1-accumulation due to defective autophagy has an effect on hepatosteatosis in response to fasting and was concerned about the possibility that mitochondrial dysfunction due to loss of *Atg7* or *Atg5* simply causes defective β -oxidation. To respond those comments, we conducted oil-red O staining and electron microscopic

analysis. As shown in Supplementary Figure S9 in the revised manuscript, physiological hepatosteatosis upon fasting was suppressed by loss of *Atg7*, due to decreased expression of genes encoding enzymes related to lipogenesis (*i.e.*, LXR α targets) (Supplementary Figure S8a). Microscopic analysis showed morphologically normal mitochondria in hepatocytes of both 4-week-old *Atg7^{ff};Alb-Cre* mice and of *Atg5^{ff};Mx1-Cre* mice at 1-2 weeks after injection of pIpC (Supplementary Figure S3 of the revised manuscript), suggesting that mitochondrial function in the mutant mouse livers is intact at this stage. We described these results in the revised manuscript (line 139-141 and line 367-369 in the revised manuscript).

Comment-1

Given the possibility that the observed reduction in hepatic lipid oxidation in liver ATG7 KO mice might arise from defective mitochondrial autophagy and mitochondrial dysfunction, it would be helpful to examine the functional changes of mitochondria (e.g. by electron microscopy, etc) in relation to NCoR1 regulation.

Reply-1

Thank you for the comment. As described in “Reply of general comments”, we performed microscopic analysis and observed morphologically normal mitochondria in hepatocytes of both 4-week-old *Atg7^{ff};Alb-Cre* mice and of *Atg5^{ff};Mx1-Cre* mice at 1-2 weeks after injection of pIpC (Supplementary Figure S3 of the revised manuscript), implying that mitochondrial function in the mutant mouse livers is intact at this stage. We described the result in the revised manuscript (line 139-141 in the revised manuscript).

Comment-2

In addition to the immunofluorescent results in Fig 5e, more evidence is needed to support autophagy-mediated degradation of NCoR1, e.g. the co-localization of NCoR1/GABARAP puncta with a lysosome marker such as LAMP2 in response to fasting, or additional markers and changes in their co-localization with NCoR1 in LC3-GFP-expressing cells, or even with endogenous LC3 co-localization if technically feasible. Immunogold electron microscopy with anti-NCoR1 antibody may be another useful strategy.

Reply-2

In addition to double-immunostaining with anti-GABARAP and anti-NCoR1 antibodies, we carried out immunofluorescent analysis with anti-LAMP1 and anti-NCoR1 antibodies. As expected, both proteins were co-localized in wild-type but not *Atg7*-deficient HepG2 cells (Figure 5e in the revised manuscript). Immunoelectron microscopic analysis with anti-NCoR1 antibody is critical for proving the localization of NCoR1 on the isolation membrane and/or autophagosome, but technically difficult at present.

Comment-3

*Are there differences in lipid contents or fatty acid oxidation levels between *Atg7-p62* double KO and *Atg7-p62-NCoR1* triple KO livers under fed versus fasted states?*

Reply-3

While the level of blood ketone body in *Atg7^{ff};p62^{ff};Alb-Cre* mice under fasting condition was lower than that in control mice, the level in *Atg7^{ff};p62^{ff};NCoR1^{ff};Alb-Cre* mice was comparable with that in control mice (Fig. 4d in the revised manuscript), suggesting the accumulation of

NCoR1 is a primary cause for the impairment of β -oxidation (line 247-249 in the revised manuscript).

Comment-4

In Fig 2b, it appears that nuclear NCoR1 levels dramatically decreased upon fasting in liver ATG7-deficient mice, but this did not happen in liver ATG7-p62 double KO mice in Fig 4b. Does this suggest p62 has a role in regulating NCoR1 during fasting in the absence of autophagy? How come p62 protein was not detectable in the control mice in Fig 4b? Fasting induction of PPAR α was not consistently observed in these different models. All immunoblots from multiple replicates of the experiments should be quantified to show statistical significance.

Reply-4

We cannot exclude a possibility that p62 has a role on NCoR1 regulation. But, in cell-based analysis, we did not observe any effect of p62-knockdown on both nuclear and cytoplasmic NCoR1 levels (Figure 3d in the revised manuscript). The down-regulation of NCoR1 protein under fasting might be attributed to severe liver injury recognized in liver specific *Atg7*-deficient mice, which is restored by additional loss of *p62* (Supplementary Figure S5a and b). In Figure 4b, when overexposed, we detected p62 protein in control livers. We replaced the blot with representative one (Figure 4b in the revised manuscript). We quantified all immunoblot analyses, and the quantification revealed that in any genotype mice utilized in this manuscript, fasting does not increase the level of PPAR α protein statistically.

Comment-5

For immunostaining of NCoR1 in Fig 2c and 4c, the quality of fluorescent images can be improved, perhaps using nuclear markers with color images.

Reply-5

According to this suggestion together with comment 6 of the Reviewer 1, we retried the immunofluorescence analysis. As shown in Figure 2e and 4c in the revised manuscript, the quality was significantly improved.

Comment-6

For Supplementary Table S1, the data might be better presented if changes were shown as heat maps.

Reply-6

In accordance with this comment, we showed data of lipidome analysis as heat maps (Supplementary Figure S1 in the revised manuscript).

Reviewer #3 (Remarks to the Author):

General comments

*The authors identified a novel autophagy-dependent mechanism that control lipid homeostasis. This study started after a lipidomic analyses on liver of conditional *Atg7* knockout that showed an accumulation of Acylcarnitine species. This finding together with a decrease of ketone bodies in the blood suggest an impairment in lipid oxidation. This was the rationale to look at PPAR α , a critical nuclear receptor that controls lipid metabolism and found that was decreased because the inhibitor NCoR1 was dramatically upregulated. Gain and loss of function experiments in vivo and in vitro*

including also rescue experiments confirmed that NCoR was indeed a substrate of autophagy and was the culprit of PPAR α inhibition and of the lipid metabolism reduction. Finally, they also found a LIR/GIR that allows the NCoR binding to Gabarap protein on the autophagosome. The paper is well written the experiments are well designed and data are consistent with author hypothesis and conclusion. This is an excellent paper with many in vivo experiments and transgenic/knockout mice. The story is novel and would clarify several missing points and contradictory data on autophagy and regulation of lipids. I have only two minor points.

Reply

We thank the reviewer for the positive evaluation of our manuscript and for the helpful comments.

Comment-1

In nuclear and cytosolic fraction western blots, actin should be checked in nuclear and Lamin in cytosolic fraction to show that there is no contamination.

Reply-1

We checked all fractionation of nucleus and cytoplasm by immunoblot analysis with anti-Gapdh (cytosolic marker) and anti-Lamin B (nuclear marker) antibodies.

Comment-2

Does the NCoR/Gabarab colocalization increase after lysosomal inhibition (BafA)? And which is the explanation of the increase Gabarap/NCoR colocalization when ATG7 is absent? The expectation is a reduction of the colocalization when the conjugation system is blocked.

Reply-2

Thank you for the comment. We performed double-immunostaining with anti-GABARAP and anti-NCoR1 antibodies under the treatment of Bafilomycin A₁ and verified the increase of GABARAP- and NCoR1-positive structures (Supplementary Figure S7). Regarding the colocalization of GABARAP with NCoR1 in *Atg7*-deficient HepG2 cells, excessive accumulated NCoR1 might sequester GABARAP due to their physical interaction. Our previous genetic studies showed formation of LC3-dots even in *Atg7*-deficient mouse hepatocytes, through the interaction of LC3 with p62 (Komatsu M et al., *J Cell Biol* 169, 425-434, 2005 and *Cell* 131, 1149-1163, 2007). We explained the above-mentioned facts in the results section (line 283-286 in the revised manuscript).

Comment-3

The authors showed in Fig2a a decrease in PPAR α transcript, is this repression consequent to NCoR upregulation or is independent of NCoR.

Reply-3

Transcription of *PPAR α* gene is regulated by PPAR α protein, and therefore this repression seems to be an NCoR1-dependent manner. In fact, in *Atg7 p62 NCoR1*-triple but not *Atg7 p62*-double knockout mouse livers, the expression of *PPAR α* was restored (Figure 4a in the revised manuscript).

REVIEWERS' COMMENTS:

Reviewer #1 (Remarks to the Author):

The authors have responded well to the comments provided.
It is interesting that the phenotype may be due to a combination of NCoR1 increase with PPAR α decrease

Reviewer #2 (Remarks to the Author):

This manuscript reports a well-conceived study showing that autophagy could act through NCoR1 to regulate hepatic lipid metabolism. The experiments were well conducted, and the results are clearly presented. The reported mechanistic findings will further our molecular understanding of the metabolic role of autophagy-related control of metabolic pathways in hepatic lipid homeostasis. The revised manuscript has included a number of additional experiments to address carefully the issues and concerns from the reviewer, and the current version is well improved in terms of corroborating the conclusions.

Reviewer #3 (Remarks to the Author):

The authors addressed my concerns

Reviewers' Comments: (italicized)

Comments

Reviewer #1 (Remarks to the Author):

The authors have responded well to the comments provided. It is interesting that the phenotype may be due to a combination of NCoR1 increase with PPARα decrease.

Reviewer #2 (Remarks to the Author):

This manuscript reports a well-conceived study showing that autophagy could act through NCoR1 to regulate hepatic lipid metabolism. The experiments were well conducted, and the results are clearly presented. The reported mechanistic findings will further our molecular understanding of the metabolic role of autophagy-related control of metabolic pathways in hepatic lipid homeostasis. The revised manuscript has included a number of additional experiments to address carefully the issues and concerns from the reviewer, and the current version is well improved in terms of corroborating the conclusions.

Reviewer #3 (Remarks to the Author):

The authors addressed my concerns.

Reply

I thank all reviewers for the positive evaluation of our revised manuscript.